# Grounding Physical Concepts of Objects and Events Through Dynamic Visual Reasoning

**Zhenfang Chen**
The University of Hong Kong

**Jiayuan Mao**
MIT CSAIL

**Jiajun Wu**
Stanford University

**Kwan-Yee K. Wong**
The University of Hong Kong

**Joshua B. Tenenbaum**
MIT BCS, CBMM, CSAIL

**Chuang Gan**
MIT-IBM Watson AI Lab

## Abstract

We study the problem of dynamic visual reasoning on raw videos. This is a challenging problem; currently, state-of-the-art models often require dense supervision on physical object properties and events from simulation, which are impractical to obtain in real life. In this paper, we present the Dynamic Concept Learner (DCL), a unified framework that grounds physical objects and events from dynamic scenes and language. DCL first adopts a trajectory extractor to track each object over time and to represent it as a latent, object-centric feature vector. Building upon this object-centric representation, DCL learns to approximate the dynamic interaction among objects using graph networks. DCL further incorporates a semantic parser to parse question into semantic programs and, finally, a program executor to run the program to answer the question, levering the learned dynamics model. After training, DCL can detect and associate objects across the frames, ground visual properties and physical events, understand the causal relationship between events, make future and counterfactual predictions, and leverage these extracted presentations for answering queries. DCL achieves state-of-the-art performance on CLEVRER, a challenging causal video reasoning dataset, even without using ground-truth attributes and collision labels from simulations for training. We further test DCL on a newly proposed video-retrieval and event localization dataset derived from CLEVRER, showing its strong generalization capacity.

## 1 Introduction

Visual reasoning in dynamic scenes involves both the understanding of compositional properties, relationships, and events of objects, and the inference and prediction of their temporal and causal structures. As depicted in Fig. 1, to answer the question *"What will happen next?"* based on the observed video frames, one needs to detect the object trajectories, predict their dynamics, analyze the temporal structures, and ground visual objects and events to get the answer *"The blue sphere and the yellow object collide"*.

Recently, various end-to-end neural network-based approaches have been proposed for joint understanding of video and language (Lei et al., 2018; Fan et al., 2019). While these methods have shown great success in learning to recognize visually complex concepts, such as human activities (Xu et al., 2017; Ye et al., 2017), they typically fail on benchmarks that require the understanding of compositional and causal structures in the videos and text (Yi et al., 2020). Another line of research has been focusing on building modular neural networks that can represent the compositional structures in scenes and questions, such as object-centric scene structures and multi-hop reasoning (Andreas et al., 2016; Johnson et al., 2017b; Hudson & Manning, 2019). However, these methods are designed for static images and do not handle the temporal and causal structure in dynamic scenes well, leading to inferior performance on video causal reasoning benchmark CLEVRER (Yi et al., 2020).

To model the temporal and causal structures in dynamic scenes, Yi et al. (2020) proposed an oracle model to combine symbolic representation with video dynamics modeling and achieved state-of-the-art performance on CLEVRER. However, this model requires videos with dense annotations for visual attributes and physical events, which are impractical or extremely labor-intensive in real scenes.

---

Project page: http://dcl.csail.mit.edu

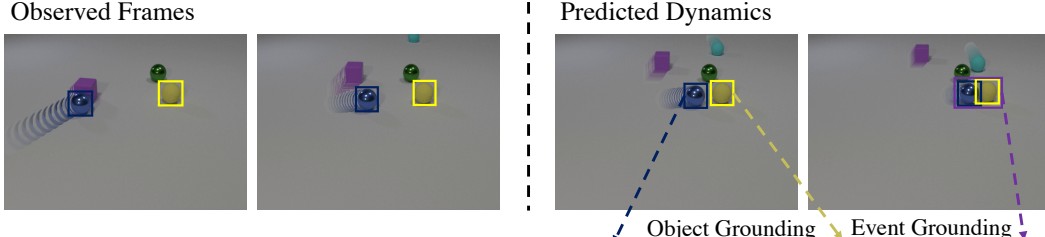

Figure 1: The process to handle visual reasoning in dynamic scenes. The trajectories of the target blue and yellow spheres are marked by the sequences of bounding boxes. Object attributes of the blue sphere and yellow sphere and the *collision* event are marked by blue, yellow and purple colors. Stroboscopic imaging is applied for motion visualization.

We argue that such dense explicit video annotations are unnecessary for video reasoning, since they are naturally encoded in the question answer pairs associated with the videos. For example, the question answer pair and the video in Fig. 1 can implicitly inform a model what the concepts "*sphere*", "*blue*", "*yellow*" and "*collide*" really mean. However, a video may contain multiple fast-moving occluded objects and complex object interactions, and the questions and answers have diverse forms. It remains an open and challenging problem to simultaneously represent objects over time, train an accurate dynamic model from raw videos, and align objects with visual properties and events for accurate temporal and causal reasoning, using vision and language as the only supervision.

Our main ideas are to factorize video perception and reasoning into several modules: object tracking, object and event concept grounding, and dynamics prediction. We first detect objects in the video, associating them into object tracks across the frames. We can then ground various object and event concepts from language, train a dynamic model on top of object tracks for future and counterfactual predictions, analyze relationships between events, and answer queries based on these extracted representations. All these modules can be trained jointly by watching videos and reading paired questions and answers.

To achieve this goal, we introduce Dynamic Concept Learner (DCL), a unified neural-symbolic framework for recognizing objects and events in videos and analyzing their temporal and causal structures, without explicit annotations on visual attributes and physical events such as collisions during training. To facilitate model training, a multi-step training paradigm has been proposed. We first run an object detector on individual frames and associate objects across frames based on a motion-based correspondence. Next, our model learns concepts about object properties, relationships, and events by reading paired questions and answers that describe or explain the events in the video. Then, we leverage the acquired visual concepts in the previous steps to refine the object association across frames. Finally, we train a dynamics prediction network (Li et al., 2019b) based on the refined object trajectories and optimize it jointly with other learning parts in this unified framework. Such a training paradigm ensures that all neural modules share the same latent space for representing concepts and they can bootstrap the learning of each other.

We evaluate DCL's performance on CLEVRER, a video reasoning benchmark that includes descriptive, explanatory, predictive, and counterfactual reasoning with a uniform language interface. DCL achieves state-of-the-art performance on all question categories and requires no scene supervision such as object properties and collision events. To further examine the grounding accuracy and transferability of the acquired concepts, we introduce two new benchmarks for video-text retrieval and spatial-temporal grounding and localization on the CLEVRER videos, namely CLEVRER-Retrieval and CLEVRER-Grounding. Without any further training, our model generalizes well to these benchmarks, surpassing the baseline by a noticeable margin.

## 2 RELATED WORK

Our work is related to reasoning and answering questions about visual content. Early studies like (Wu et al., 2016; Zhu et al., 2016; Gan et al., 2017) typically adopted monolithic network architectures and mainly focused on visual understanding. To perform deeper visual reasoning, neural module networks were extensively studied in recent works (Johnson et al., 2017a; Hu et al., 2018; Hudson & Manning, 2018; Amizadeh et al., 2020), where they represent symbolic operations with small neural networks and perform multi-hop reasoning. Some previous research has also attempted to learn

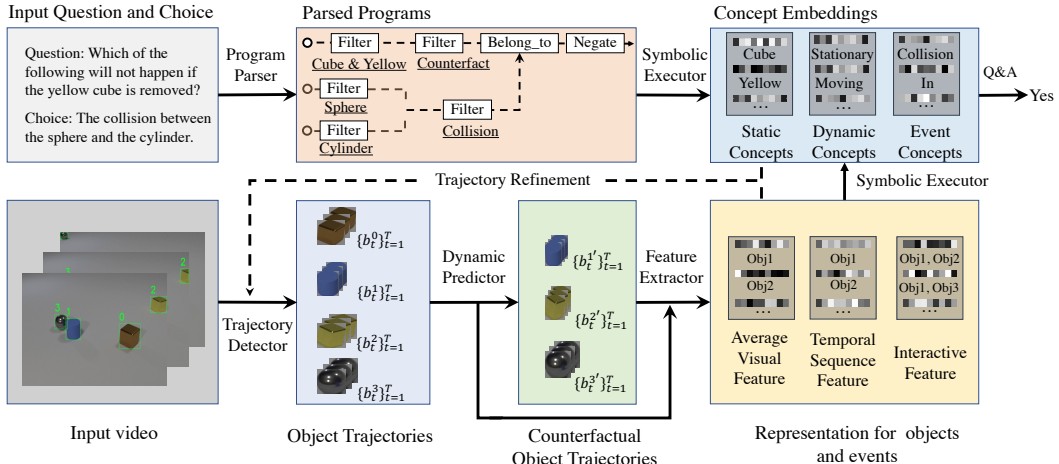

Figure 2: DCL's architecture for counterfactual questions during inference. Given an input video and its corresponding question and choice, we first use a program parser to parse the question and the choice into executable programs. We adopt an object trajectory detector to detect trajectories of all objects. Then, the extracted objects are sent to a dynamic predictor to predict their dynamics. Next, the extracted objects are sent to the feature extractor to extract latent representations for objects and events. Finally, we feed the parsed programs and latent representation to the symbolic executor to answer the question and optimize concept learning.

visual concepts through visual question answering (Mao et al., 2019). However, it mainly focused on learning static concepts in images, while our DCL aims at learning dynamic concepts like *moving* and *collision* in videos and at making use of these concepts for temporal and causal reasoning.

Later, visual reasoning was extended to more complex dynamic videos (Lei et al., 2018; Fan et al., 2019; Li et al., 2020; 2019a; Huang et al., 2020). Recently, Yi et al. (2020) proposed CLEVRER, a new video reasoning benchmark for evaluating computational models' comprehension of the causal structure behind physical object interaction. They also developed an oracle model, combining the neuro-symbolic visual question-answering model (Yi et al., 2018) with the dynamics prediction model (Li et al., 2019b), showing competitive performance. However, this model requires explicit labels for object attributes, masks, and spatio-temporal localization of events during training. Instead, our DCL has no reliance on any labels for objects and events and can learn these concepts through natural supervision (i.e., videos and question-answer pairs).

Our work is also related to temporal and relational reasoning in videos via neural networks (Wang & Gupta, 2018; Materzynska et al., 2020; Ji et al., 2020). These works typically rely on specific action annotations, while our DCL learns to ground object and event concepts and analyze their temporal relations through question answering. Recently, various benchmarks (Riochet et al., 2018; Bakhtin et al., 2019; Girdhar & Ramanan, 2020; Baradel et al., 2020; Gan et al., 2020) have been proposed to study dynamics and reasoning in physical scenes. However, these datasets mainly target at pure video understanding and do not contain natural language question answering. Much research has been studying dynamic modeling for physical scenes (Lerer et al., 2016; Battaglia et al., 2013; Mottaghi et al., 2016; Finn et al., 2016; Shao et al., 2014; Fire & Zhu, 2015; Ye et al., 2018; Li et al., 2019b). We adopt PropNet (Li et al., 2019b) for dynamics prediction and feed the predicted scenes to the video feature extractor and the neuro-symbolic executor for event prediction and question answering.

While many works (Zhou et al., 2019; 2018; Gan et al., 2015) have been studying on the problems of understanding human actions and activities (e.g., running, cooking, cleaning) in videos, our work's primary goal is to design a unified framework for learning physical object and event concepts (e.g., collision, falling, stability). These tasks are of great importance in practical applications such as industrial robot manipulation which requires AI systems with human-like physical common sense.

## 3 DYNAMIC CONCEPT LEARNER

In this section, we introduce a new video reasoning model, Dynamic Concept Learner (DCL), which learns to recognize video attributes, events, and dynamics and to analyze their temporal and causal structures, all through watching videos and answering corresponding questions. DCL contains five modules, 1) an object trajectory detector, 2) video feature extractor, 3) a dynamic predictor, 4) a language program parser, and 5) a neural symbolic executor. As shown in Fig. 2, given an input

video, the trajectory detector detects objects in each frame and associates them into trajectories; the feature extractor then represents them as latent feature vectors. After that, DCL quantizes the objects' static concepts (i.e., color, shape, and material) by matching the latent object features with the corresponding concept embeddings in the executor. As these static concepts are motion-independent, they can be adopted as an additional criteria to refine the object trajectories. Based on the refined trajectories, the dynamics predictor predicts the objects' movement and interactions in future and counterfactual scenes. The language parser parses the question and choices into functional programs, which are executed by the program executor on the latent representation space to get answers.

The object and event concept embeddings and the object-centric representation share the same latent space; answering questions associated with videos can directly optimize them through backpropagation. The object trajectories and dynamics can be refined by the object static attributes predicted by DCL. Our framework enjoys the advantages of both transparency and efficiency, since it enables step-by-step investigations of the whole reasoning process and has no requirements for explicit annotations of visual attributes, events, and object masks.

## 3.1 MODEL DETAILS

**Object Detection and Tracking.** Given a video, the object trajectory detector detects object proposals in each frame and connects them into object trajectories $O = \{o^n\}_{n=1}^N$, where $o^n = \{b_t^n\}_{t=1}^T$ and $N$ is the number of objects in the video. $b_t = [x_t^n, y_t^n, w_t^n, h_t^n]$ is an object proposal at frame $t$ and $T$ is the frame number, where $(x_t^n, y_t^n)$ denotes the normalized proposal coordinate center and $w_t^n$ and $h_t^n$ denote the normalized width and height, respectively.

The object detector first uses a pre-trained region proposal network (Ren et al., 2015) to generate object proposals in all frames, which are further linked across connective frames to get all objects' trajectories. Let $\{b_t^i\}_{i=1}^N$ and $\{b_{t+1}^j\}_{j=1}^N$ to be two sets of proposals in two connective frames. Inspired by Gkioxari & Malik (2015), we define a connection score $s_l$ between $b_t^i$ and $b_{t+1}^j$ to be

$$s_l(b_t^i, b_{t+1}^j) = s_c(b_t^i) + s_c(b_{t+1}^j) + \lambda_1 \cdot IoU(b_t^i, b_{t+1}^j), \tag{1}$$

where $s_c(b_t^i)$ is the confidence score of the proposal $b_t^i$, $IoU$ is the intersection over union and $\lambda_1$ is a scalar. Gkioxari & Malik (2015) adopts a greedy algorithm to connect the proposals without global optimization. Instead, we assign boxes $\{b_{t+1}^j\}_{j=1}^N$ at the $t+1$ frame to $\{b_t^i\}_{i=1}^N$ by a linear sum assignment.

**Video Feature Extraction.** Given an input video and its detected object trajectories, we extract three kinds of latent features for grounding object and event concepts. It includes 1) the average visual feature $f^v \in \mathbb{R}^{N \times D_1}$ for static attribute prediction, 2) temporal sequence feature $f^s \in \mathbb{R}^{N \times 4T}$ for dynamic attribute and unary event prediction, and 3) interactive feature $f^c \in \mathbb{R}^{K \times N \times N \times D_2}$ for *collision* event prediction, where $D_1$ and $D_2$ denote dimensions of the features and $K$ is the number of sampled frames. We give more details on how to extract these features in Appendix B.

**Grounding Object and Event Concepts.** Video Reasoning requires a model to ground object and event concepts in videos. DCL achieves this by matching object and event representation with object and event embeddings in the symbolic executor. Specifically, DCL calculates the confidence score that the $n$-th object is *moving* by $\left[\cos(s^{\text{moving}}, m_{da}(f_n^s)) - \delta\right]/\lambda$, where $f_n^s$ denotes the temporal sequence feature for the $n$-th object, $s^{\text{moving}}$ denotes a vector embedding for concept *moving*, and $m_{da}$ denotes a linear transformation, mapping $f_n^s$ into the dynamic concept representation space. $\delta$ and $\lambda$ are the shifting and scaling scalars, and $\cos()$ calculates the cosine similarity between two vectors. DCL grounds static attributes and the collision event similarly, matching average visual features and interactive features with their corresponding concept embeddings in the latent space. We give more details on the concept and event quantization in Appendix E.

**Trajectory Refinement.** The connection score in Eq. 1 ensures the continuity of the detected object trajectories. However, it does not consider the objects' visual appearance; therefore, it may fail to track the objects and may connect inconsistent objects when different objects are close to each other and moving rapidly. To detect better object trajectories and to ensure the consistency of visual appearance along the track, we add a new term to Eq. 1 and re-define the connection score to be

$$s_l(\{b_m^i\}_{m=0}^t, b_{t+1}^j) = s_c(b_t^i) + s_c(b_{t+1}^j) + \lambda_1 \cdot IoU(b_t^i, b_{t+1}^j) + \lambda_2 \cdot f_{\text{appear}}(\{b_m^i\}_{m=0}^t, b_{t+1}^j), \tag{2}$$

where $f_{\text{appear}}(\{b_m^i\}_{m=0}^t, b_{t+1}^j)$ measures the attribute similarity between the newly added proposal $b_{t+1}^j$ and all proposals $(\{b_m^i\}_{m=0}^t$ in previous frames. We define $f_{\text{appear}}$ as

$$f_{\text{appear}}(\{b_m^i\}_{m=0}^t, b_{t+1}^j) = \frac{1}{3 \times t} \sum_{attr} \sum_{m=0}^t f_{\text{attr}}(b_m^i, b_{t+1}^j), \quad (3)$$

where attr $\in \{$color, material, shape$\}$. $f_{\text{attr}}(b_m, b_{t+1})$ equals to 1 when $b_m^i$ and $b_{t+1}^j$ have the same attribute, and 0 otherwise. In Eq. 2, $f_{\text{appear}}$ ensures that the detected trajectories have consistent visual appearance and helps to distinguish the correct object when different objects are close to each other in the same frame. These additional static attributes, including color, material, and shape, are extracted without explicit annotation during training. Specifically, we quantize the attributes by choosing the concept whose concept embedding has the best cosine similarity with the object feature. We iteratively connect proposals at the $t+1$ frame to proposals at the $t$ frame and get a set of object trajectories $O = \{o^n\}_{n=1}^N$, where $o^n = \{b_t^n\}_{t=1}^T$.

**Dynamic Prediction.** Given an input video and the refined trajectories of objects, we predict the locations and RGB patches of the objects in future or counterfactual scenes with a Propagation Network (Li et al., 2019b). We then generate the predicted scenes by pasting RGB patches into the predicted locations. The generated scenes are fed to the feature extractor to extract the corresponding features. Such a design enables the question answer pairs associated with the predicted scenes to optimize the concept embeddings and requires no explicit labels for collision prediction, leading to better optimization. This is different from Yi et al. (2020), which requires dense collision event labels to train a collision classifier.

To predict the locations and RGB patches, the dynamic predictor maintains a directed graph $\langle V, D \rangle = \left\langle \{v_n\}_{n=1}^N, \{d_{n_1,n_2}\}_{n_1=1,n_2=1}^{N,N} \right\rangle$. The $n$-th vertex $v_n$ is represented by a concatenation of tuple $\langle b_t^n, p_t^n \rangle$ over a small time window $w$, where $b_t^n = [x_t^n, y_t^n, w_t^n, h_t^n]$ is the $n$-th object's normalized coordinates and $p_t^n$ is a cropped RGB patch centering at $(x_t^n, y_t^n)$. The edge $d_{n_1,n_2}$ denotes the relation between the $n_1$-th and $n_2$-th objects and is represented by the concatenation of the normalized coordinate difference $b_t^{n_1} - b_t^{n_2}$. The dynamic predictor performs multi-step message passing to simulate instantaneous propagation effects.

During inference, the dynamics predictor predicts the locations and patches at frame $k+1$ using the features of the last $w$ observed frames in the original video. We get the predictions at frame $k+2$ by autoregressively feeding the predicted results at frame $k+1$ as the input to the predictor. To get the counterfactual scenes where the $n$-th object is removed, we remove the $n$-th vertex and its associated edges from the input to predict counterfactual dynamics. Iteratively, we get the predicted normalized coordinates $\{\hat{b}_{k'}^n\}_{n=1,k'=1}^{N,K'}$ and RGB patches $\{\hat{p}_{k'}^n\}_{n=1,k'=1}^{N,K}$ at all predicted $K'$ frames. We give more details on the dynamic predictor at Appendix C.

**Language Program Parsing.** The language program parser aims to translate the questions and choices into executable symbolic programs. Each executable program consists of a series of operations like selecting objects with certain properties, filtering events happening at a specific moment, finding the causes of an event, and eventually enabling transparent and step-by-step visual reasoning. Moreover, these operations are compositional and can be combined to represent questions with various compositionality and complexity. We adopt a seq2seq model (Bahdanau et al., 2015) with an attention mechanism to translate word sequences into a set of symbolic programs and treat questions and choices, separately. We give detailed implementation of the program parser in Appendix D.

**Symbolic Execution.** Given a parsed program, the symbolic executor explicitly runs it on the latent features extracted from the observed and predicted scenes to answer the question. The executor consists of a series of functional modules to realize the operators in symbolic programs. The last operator's output is the answer to the question. Similar to Mao et al. (2019), we represent all object states, events, and results of all operators in a probabilistic manner during training. This makes the whole execution process differential w.r.t. the latent representations from the observed and predicted scenes. It enables the optimization of the feature extractor and concept embeddings in the symbolic executor. We provide the implementation of all the operators in Appendix E.

## 3.2 TRAINING AND INFERENCE

**Training.** We follow a multi-step training paradigm to optimize the model: 1) We first extract object trajectories with the scoring function in Eq. 1 and optimize the video feature extractor and

| Methods | Extra Labels | | Descriptive | Explanatory | | Predictive | | Counterfactual | |
|---|---|---|---|---|---|---|---|---|---|
| | Attr. | Prog. | | per opt. | per ques. | per opt. | per ques. | per opt. | per ques. |
| CNN+MLP | | | 48.4 | 54.9 | 18.3 | 50.5 | 13.2 | 55.2 | 9.0 |
| CNN+LSTM | | | 51.8 | 62.0 | 17.5 | **57.9** | 31.6 | **61.2** | **14.7** |
| Memory | No | No | 54.7 | 53.7 | 13.9 | 50.0 | **33.1**. | 54.2 | 7.0 |
| HCRN | | | 55.7 | **63.3** | **21.0** | 54.1 | 21.0 | 57.1 | 11.5 |
| MAC (V) | | | **85.6** | 59.5 | 12.5 | 51.0 | 16.5 | 54.6 | 13.7 |
| TVQA+ | Yes | No | 72.0 | 63.3 | **23.7** | **70.3** | **48.9** | 53.9 | 4.1 |
| MAC (V+) | | | **86.4** | **70.5** | 22.3 | 59.7 | 42.9 | **63.5** | **25.1** |
| IEP (V) | | | 52.8 | 52.6 | 14.5 | 50.0 | 9.7 | 53.4 | 3.8 |
| TbD-net (V) | No | Yes | 79.5 | 61.6 | 3.8 | 50.3 | 6.5 | 56.1 | 4.4 |
| DCL (Ours) | | | **90.7** | **89.6** | **82.8** | **90.5** | **82.0** | **80.4** | **46.5** |
| NS-DR | | | 88.1 | 87.6 | 79.6 | 82.9 | 68.7 | 74.1 | 42.4 |
| NS-DR (NE) | Yes | Yes | 85.8 | 85.9 | 74.3 | 75.4 | 54.1 | 76.1 | 42.0 |
| DCL-Oracle (Ours) | | | **91.4** | **89.8** | **82.0** | **90.6** | **82.1** | **80.7** | **46.9** |

Table 1: Question-answering accuracy on CLEVRER. The first and the second parts of the table show the models without and with visual attribute and event labels during training, respectively. Best performance is highlighted in boldface. DCL and DCL-Oracle denote our models trained without and with labels of visual attributes and events, respectively.

concept embeddings in the symbolic executor with only descriptive and explanatory questions; 2) We quantize the static attributes for all objects with the feature extractor and the concept embeddings learned in Step 1) and refine object trajectories with the scoring function Eq. 2; 3) Based on the refined trajectories, we train the dynamic predictor and predict dynamics for future and counterfactual scenes; 4) We train the full DCL with all the question answer pairs and get the final model. The program executor is fully differentiable w.r.t. the feature extractor and concept embeddings. We use cross-entropy loss to supervise open-ended questions and use mean square error loss to supervise counting questions. We provide specific loss functions for each module in Appendix H.

**Inference.** During inference, given an input video and a question, we first detect the object trajectories and predict their motions and interactions in future and counterfactual scenes. We then extract object and event features for both the observed and predicted scenes with the feature extractor. We parse the questions and choices into executable symbolic programs. We finally execute the programs on the latent feature space and get the answer to the question.

## 4 EXPERIMENTS

To show the proposed DCL's advantages, we conduct extensive experiments on the video reasoning benchmark CLEVRER. Existing other video datasets either ask questions about the complex visual context (Tapaswi et al., 2016; Lei et al., 2018) or study dynamics and reasoning without question answering (Girdhar & Ramanan, 2020; Baradel et al., 2020). Thus, they are unsuitable for evaluating video causal reasoning and learning object and event concepts through question answering. We first show its strong performance on video causal reasoning. Then, we show DCL's ability on concept learning, predicting object visual attributes and events happening in videos. We show DCL's generalization capacity to new applications, including CLEVRER-Grounding and CLEVRER-Retrieval. We finally extend DCL to a real block tower video dataset (Lerer et al., 2016).

### 4.1 IMPLEMENTATION DETAILS

Following the experimental setting in Yi et al. (2020), we train the language program parser with 1000 programs for all question types. We train all our models without attribute and event labels. Our models for video question answering are trained on the training set, tuned on the validation set, and evaluated in the test set. To show DCL's generalization capacity, we build CLEVRER-Grounding and CLEVRER-Retrieval datasets from the original CLEVRER videos and their associated video annotations. We provide more implementation details in Appendix A.

### 4.2 COMPARISONS ON TEMPORAL AND CAUSAL REASONING

We compare our DCL with previous methods on CLEVRER, including **Memory** (Fan et al., 2019), **IEP** (Johnson et al., 2017b), **TbD-net** (Mascharka et al., 2018), **TVQA+** (Lei et al., 2018), **NS-**

| Methods | Static Attributes | | | Dynamic Attributes | | Events | | |
|---------|-------|-------|----------|--------|------------|------|------|-----------|
|         | Color | Shape | Material | Moving | Stationary | In   | Out  | Collision |
| DCL     | 99.7  | 99.2  | 99.6     | 89.7   | 93.3       | 99.2 | 98.9 | 96.9      |

Table 2: Evaluation of video concept learning on the validation set.

**DR** (Yi et al., 2020), **MAC (V)** (Hudson & Manning, 2018) and its attribute-aware variant, **MAC (V+)**. We refer interested readers to CLEVRER (Yi et al., 2020) for more details. Additionally, we also include a recent state-of-the-art VQA model **HCRN** (Le et al., 2020) for performance comparison, which adopts a conditional relation network for representation and reasoning over videos. To provide more extensive analysis, we introduce DCL-Oracle by adding object attribute and collision supervisions into DCL's training. We summarize their requirement for visual labels and language programs in the second and third columns of Table 1.

According to the results in Table 1, we have the following observations. Although **HCRN** achieves state-of-the-art performance on human-centric action datasets (Jang et al., 2017; Xu et al., 2017; 2016), it only performs slightly better than **Memory** and much worse than NS-DR on CLEVRER. We believe the reason is that **HCRN** mainly focuses on motion modeling across frames while CLEVRER requires models to perform dynamic visual reasoning on videos and analyze its temporal and causal structures. NS-DR performs best among all the baseline models, showing the power of combining symbolic representation with dynamics modeling. Our model achieves the state-of-the-art question answering performance on all kinds of questions even without visual attributes and event labels from simulations during training, showing its effectiveness and label-efficiency. Compared with NS-DR, our model achieves more significant gains on predictive and counterfactual questions than that on the descriptive questions. This shows DCL's effectiveness in modeling for temporal and causal reasoning. Unlike NS-DR, which directly predicts collision event labels with its dynamic model, DCL quantizes concepts and executes symbolic programs in an end-to-end training manner, leading to better predictions for dynamic concepts. DCL-Oracle shows the upper-bound performance of the proposed model to ground physical object and event concepts through question answering.

### 4.3 EVALUATION OF OBJECT AND EVENT CONCEPT GROUNDING IN VIDEOS

Previous methods like MAC (V) and TbD-net (V) did not learn explicit concepts during training, and NS-DR required intrinsic attribute and event labels as input. Instead, DCL can directly quantize video concepts, including static visual attributes (i.e. *color*, *shape*, and *material*), dynamic attributes (i.e. *moving* and *stationary*) and events (i.e. *in*, *out*, and *collision*). Specifically, DCL quantizes the concepts by mapping the latent object features into the concept space by linear transformation and calculating their cosine similarities with the concept embeddings in the neural-symbolic executor.

We predict the static attributes of each object by averaging the visual object features at each sampled frame. We regard an object to be *moving* if it moves at any frame, and otherwise *stationary*. We consider there is a *collision* happening between a pair of objects if they collide at any frame of the video. We get the ground-truth labels from the provided video annotation and report the accuracy in table 2 on the validation set.

We observe that DCL can learn to recognize different kinds of concepts without explicit concept labels during training. This shows DCL's effectiveness to learn object and event concepts through natural question answering. We also find that DCL recognizes static attributes and events better than dynamic attributes. We further find that DCL may misclassify objects to be "stationary" if they are missing for most frames and only move slowly at specific frames. We suspect the reason is that we only learn the dynamic attributes through question answering and question answering pairs for such slow-moving objects rarely appear in the training set.

### 4.4 GENERALIZATION

We further apply DCL to two new applications, including **CLEVRER-Grounding**, spatio-temporal localization of objects or events in a video, and **CLEVRER-Retrieval**, finding semantic-related videos for the query expressions and vice versa.

We first build datasets for video grounding and video-text retrieval by synthesizing language expressions for videos in CLEVRER. We generate the expressions by filling visual contents from the video annotations into a set of pre-defined templates. For example, given the text template, "The <static_attribute> that is <dynamic_attribute> <time_identifier>", we can fill it and generate "*The*

**Query:** *The collision that happens after the blue sphere exits the scene.*

**Query:** *A video that contains a gray metal cube that enters the scene.*

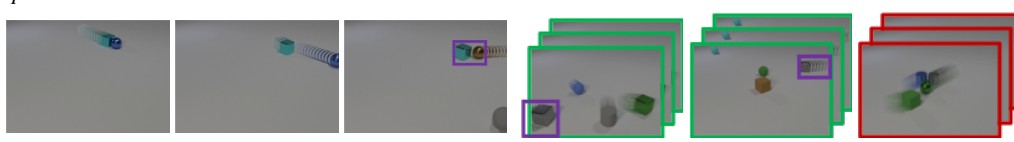

App1: CLEVRER-Grounding.

App2: CLEVRER-Retrieval.

Figure 3: Examples of CLEVRER-Grounding and CLEVRER-Retrieval Datasets. The target region are marked by purple boxes and stroboscopic imaging is applied for visualization purposes. In CLEVRER-Retrieval, we mark randomly-selected positive and negative gallery videos with green and red borders, respectively.

| Methods | Spatial Acc.↑ | | Spatial mIoU.↑ | | Frame Diff.↓ | | |
|---|---|---|---|---|---|---|---|
| | Obj. | Col. | Obj. | Col. | In | Out | Col. |
| WSSTG | 34.4 | 10.3 | 34.9 | 15.6 | 21.4 | 50.4 | 37.5 |
| DCL | **91.9** | **88.3** | **90.0** | **79.0** | **5.5** | **4.4** | **4.5** |

Table 3: Evaluation of video grounding. For spatial grounding, we consider it to be accurate if the IoU between the detected trajectory and the ground-truth trajectory is greater than 0.5.

| Methods | Text-to-Video↑ | | | | Video-to-Text↑ |
|---|---|---|---|---|---|
| | Obj. | In | Out | Col. | |
| WSSTG | 2.2 | 1.3 | 3.4 | 3.3 | 7.7 |
| HGR | 16.9 | 17.2 | 18.7 | 22.2 | 15.5 |
| DCL | **73.1** | **81.9** | **88.5** | **85.4** | **78.6** |

Table 4: Evaluation of CLEVRER-Retrieval. Mean average precision (mAP) is adopted as the metric.

*metal cube that is moving when the video ends.*". Fig. 3 shows examples for the generated datasets, and we provide more statistics and examples in Appendix G. We transform the grounding and retrieval expressions into executable programs by training new language parsers on the expressions of the synthetic training set. To provide more extensive comparisons, we adopt the representative video grounding/ retrieval model WSSTG (Chen et al., 2019) as a baseline. We provide more details of the baseline implementation in Appendix A.

**CLEVRER-Grounding.** CLEVRER-Grounding contains object grounding and event grounding. For video object grounding, we localize each described object's whole trajectory and compute the mean intersection over union (IoU) with the ground-truth trajectory. For event grounding, including *collision*, *in* and *out*, we temporally localize the frame that the event happens at and calculate the frame difference with the ground-truth frames. For *collision* event, we also spatially localize the collided objects' the union box and compute it's IoU with the ground-truth. We don't perform spatial localization for *in* and *out* events since the target object usually appears to be too small to localize at the frame it enters or leaves the scene.

Table 3 lists the results. From the table, we can find that our proposed DCL transforms to the new CLEVRER-Grounding task well and achieves high accuracy for spatial localization and low frame differences for temporal localization. On the contrary, the traditional video grounding method WSSTG performs much worse, since it mainly aligns simple visual concepts between text and images and has difficulties in modeling temporal structures and understanding the complex logic.

**CLEVRER-Retrieval.** For CLEVRER-Retrieval, an expression-video pair is considered as a positive pair if the video contains the objects and events described by the expression and otherwise negative. Given a video, we define its matching similarity with the query expression to be the maximal similarity between the query expression and all the object or event proposals. Additionally, we also introduce a recent state-of-the-art video-text retrieval model HGR (Chen et al., 2020) for performance comparison, which decomposes video-text matching into global-to-local levels and performs cross-modal matching with attention-based graph reasoning. We densely compare every possible expression-video pair and use mean average precision (mAP) as the retrieval metric.

We report the retrieval mAP in Table 4. Compared with CLEVRER-Grounding, CLEVRER-Retrieval is more challenging since it contains many more distracting objects, events and expressions. WSSTG performs worse on the retrieval setting because it does not model temporal structures and understand its logic. HGR achieves better performance than the previous baseline WSSTG since it performs hierarchical modeling for events, actions and entities. However, it performs worse than DCL since it doesn't explicitly model the temporal structures and the complex logic behind the video-text pairs in CLEVRER-Retrieval. On the other hand, DCL is much more robust since it can explicitly ground object and event concepts, analyze their relations and perform step-by-step visual reasoning.

| Methods | Question Type | | | Average |
|---|---|---|---|---|
| | Query | Exist | Count | |
| MAC (V) | 92.8 | **95.5** | 75.0 | 87.7 |
| DCL (ours) | **97.0** | 95.5 | **84.1** | **92.6** |

Table 5: QA results on the block tower dataset.

| Method | Static Color | Dynamic "*falling*" |
|---|---|---|
| DCL (ours) | 98.5 | 91.8 |

Table 6: Evaluation of concept learning on the block tower dataset. Our DCL can learn to quantize the new concept "*falling*" on real videos through QA.

**Q1:** *How many objects are falling?* **A1:** *2.*
**Q2:** *Are there any falling red objects?* **A2:** *No.*
**Q3:** *Are there any falling blue objects?* **A3:** *Yes.*

**Q1:** *What is the color of the block that is at the bottom?* **A1:** *Blue.*
**Q2:** *Are there any falling yellow objects?* **A2:** *No.*

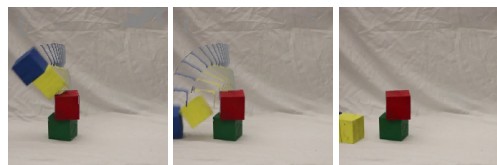

Falling block tower

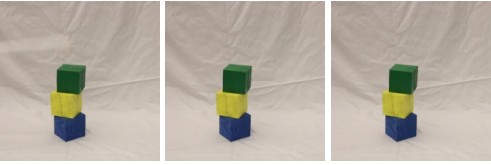

Stable block tower.

Figure 4: Typical videos and question-answer pairs of the block tower dataset. Stroboscopic imaging is applied for motion visualization.

## 4.5 EXTENSION TO REAL VIDEOS AND THE NEW CONCEPT

We further conduct experiments on a real block tower video dataset (Lerer et al., 2016) to learn the new physical concept "*falling*". The block tower dataset has 493 videos and each video contains a stable or falling block tower. Since the original dataset aims to study physical intuition and doesn't contain question-answer pairs, we manually synthesize question-answer pairs in a similar way to CLEVRER (Yi et al., 2020). We show examples of the new dataset in Fig 4. We train models on randomly-selected 393 videos and their associated question-answer pairs and evaluate their performance on the rest 100 videos.

Similar to the setting in CLEVRER, we use the average visual feature from ResNet-34 for static attribute prediction and temporal sequence feature for the prediction of the new dynamic concept "*falling*". Additionally, we train a visual reasoning baseline MAC (V) (Hudson & Manning, 2018) for performance comparison. Table 5 lists the results. Our model achieves better question-answering performance on the block tower dataset especially on the counting questions like "*How many objects are falling?*". We believe the reason is that counting questions require a model to estimate the states of each object. MAC (V) just simply adopts an MLP classifier to predict each answer's probability and doesn't model the object states. Differently, DCL answers the counting questions by accumulating the probabilities of each object and is more transparent and accurate. We also show the accuracy of color and "*falling*" concept prediction on the validation set in Table 6. Our DCL can naturally learn to ground the new dynamic concept "*falling*" in the real videos through question answering. This shows DCL's effectiveness and strong generalization capacity.

## 5 DISCUSSION AND FUTURE WORK

We present a unified neural symbolic framework, named Dynamic Concept Learner (DCL), to study temporal and causal reasoning in videos. DCL, learned by watching videos and reading question-answers, is able to track objects across different frames, ground physical object and event concepts, understand the causal relationship, make future and counterfactual predictions and combine all these abilities to perform temporal and causal reasoning. DCL achieves state-of-the-art performance on the video reasoning benchmark CLEVRER. Based on the learned object and event concepts, DCL generalizes well to spatial-temporal object and event grounding and video-text retrieval. We also extend DCL to real videos to learn new physical concepts.

Our DCL suggests several future research directions. First, it still requires further exploration for dynamic models with stronger long-term dynamic prediction capability to handle some counterfactual questions. Second, it will be interesting to extend our DCL to more general videos to build a stronger model for learning both physical concepts and human-centric action concepts.

**Acknowledgement** This work is in part supported by ONR MURI N00014-16-1-2007, the Center for Brain, Minds, and Machines (CBMM, funded by NSF STC award CCF-1231216), the Samsung Global Research Outreach (GRO) Program, Autodesk, and IBM Research.

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

## A  IMPLEMENTATION DETAILS

**DCL Implementation Details.**   Since it's extremely computation-intensive to predict the object states and events at every frame, we evenly sample 32 frames for each video. All models are trained using Adam (Kingma & Ba, 2014) for 20 epochs and the learning rate is set to $10^{-4}$. We adopt a two-stage training strategy for training the dynamic predictor. For the dynamic predictor, we set the time window size $w$, the propogation step $L$ and dimension of hidden states to be 3, 2 and 512, respectively. Following the sample rate at the observed frames, we sample a frame for prediction every 4 frames. We first train the dynamic model with only location prediction and then train it with both location and RGB patch prediction. Experimentally, we find this training strategy provides a more stable prediction. We train the language parser with the same training strategy as Yi et al. (2018) for fair comparison.

**Baseline Implementation.**   We implement the baselines HCRN (Le et al., 2020), HGR Chen et al. (2020) and WSSTG (Chen et al., 2019) carefully based on the public source code. WSSTG first generate a set of object or event candidates and match them with the query sentence. We choose the proposal candidate with the best similarity as the grounding result. For object grounding, we use the same tube trajectory candidates as we use for implementing DCL. For grounding event concepts *in* and *out*, we treat each object at each sampled frame as a potential candidate for selection. For grounding event concept *collision*, we treat the union regions of any object pairs as candidates. For CLEVRER-Retrieval, we treat the proposal candidate with the best similarity as the similarity score between the video and the query sentence. We train WSSTG with a synthetic training set generated from the videos of CLEVRER-VQA training set. A fully-supervised triplet loss is adopted to optimize the model.

## B  FEATURE EXTRACTION

We evenly sample $K$ frames for each video and use a ResNet-34 (He et al., 2016) to extract visual features. For the $n$-th object in the video, we define its average visual feature to be $f_n^v = \frac{1}{K}\sum_{k=1}^{K} f_k^n$, where $f_k^n$ is the concatenation of the regional feature and the global context feature at the $k$-th frame. We define its temporal sequence feature $f_n^s$ to be the contenation of $[x_t^n, y_t^n, w_t^n, h_t^n]$ at all $T$ frames, where $(x_t^n, y_t^n)$ denotes the normalized object coordinate centre and $w_t^n$ and $h_t^n$ denote the normalized width and height, respectively. For the collision feature between the $n_1$-th object and $n_2$-th objet at the $k$-th frame, we define it to be $f_{n_1,n_2,k}^c = f_{n_1,n_2,k}^u || f_{n_1,n_2,k}^{loc}$, where $f_{n_1,n_2,k}^u$ is the ResNet feature of the union region of the $n_1$-th and $n_2$-th objects at the $k$-th frame and $f_{n_1,n_2,k}^{loc}$ is a spatial embedding for correlations between bounding box trajectories. We define $f_{n_1,n_2,k}^{loc} = \mathrm{IoU}(s_{n_1}, s_{n_2})||(s_{n_1} - s_{n_2})||(s_{n_1} \times s_{n_2})$, which is the concatenation of the intersection over union (IoU), difference ($-$) and multiplication ($\times$) of the normalized trajectory coordinates for the $n_1$-th and $n_2$-th objects centering at the $k$-th frame. We padding $f_{n1,n2,k}^u$ with a zero vector if either the $n_1$-th or the $n_2$-th objects doesn't appear at the $k$-th frame.

## C  DYNAMIC PREDICTOR

To predict the locations and RGB patches, the dynamic predictor maintains a directed graph $\langle V, D \rangle = \left\langle \{v_n\}_{n=1}^N, \{d_{n_1,n_2}\}_{n_1=1,n_2=1}^{N,N} \right\rangle$. The $n$-th vertex $o_n$ is represented by the concatenation of its normalized coordinates $b_t^n = [x_t^n, y_t^n, w_t^n, h_t^n]$ and RGB patches $p_t^n$. The edge $d_{n_1,n_2}$ is represented by the concatenation of the normalized coordinate difference $b_t^{n_1} - b_t^{n_2}$. To capture the object dynamics, we concatenate the features over a small history window. To predict the dynamics at the $k + 1$ frame, we first encode the vertexes and edges

$$e_{n,k}^o = f_O^{enc}(||_{t=k-w}^k (b_t^n || p_t^n)), \quad e_{n_1,n_2,k}^r = f_R^{enc}(||_{t=k-w}^k (b_t^{n_1} - b_t^{n_2})), \tag{4}$$

where $||$ indicates concatenation, $w$ is the history window size, $f_O^{enc}$ and $f_R^{enc}$ are CNN-based encoders for objects and relations. $w$ is set to 3. We then update the object influences $\{h_{n,k}^l\}_{n=1}^N$ and

relation influences $\{e^l_{n_1,n_2,k}\}^{N,N}_{n_1=1,n_2=1}$ through $L$ propagation steps. Specifically, we have

$$e^l_{n_1,n_2,k} = f_R(e^r_{n_1,n_2,k}, h^{l-1}_{n_1,k}, h^{l-1}_{n_2,k}), \quad h^l_{n,k} = f_O(e^o_{n,k}, \sum_{n_1,n_2} e^l_{n_1,n_2,k}, h^{l-1}_{n,k}), \quad (5)$$

where $l \in [1, L]$, denoting the $l$-th step, $f_O$ and $f_R$ denote the object propagator and relation propagator, respectively. We initialize $h^o_{n,t} = \mathbf{0}$. We finally predict the states of objects and relations at the $k+1$ frame to be

$$\hat{b}^n_{k+1} = f^{pred}_{O_1}(e^o_{n,k}, h^L_{n,k}), \quad \hat{p}^n_{k+1} = f^{pred}_{O_2}(e^o_{n,k}, h^L_{n,k}), \quad (6)$$

where $f^{pred}_{O_1}$ and $f^{pred}_{O_2}$ are predictors for the normalized object coordinates and RGB patches at the next frame. We optimize this dynamic predictor by mimizing the $\mathcal{L}_2$ distance between the predicted $\hat{b}^n_{k+1}, \hat{p}^n_{k+1}$ and the real future locations $b^n_{k+1}$ and extracted patches $p^n_{k+1}$.

During inference, the dynamics predictor predicts the locations and patches at $k+1$ frames by using the features of the last $w$ observed frames in the original video. We get the predictions at the $k+2$ frames by feeding the predicted results at the $k+1$ frame to the encoder in Eq. 4. To get the counterfactual scenes where the $n$-th object is removed, we use the first $w$ frames of the original video as the start point and remove the $n$-th vertex and its associated edges of the input to predict counterfactual dynamics. Iteratively, we get the predicted normalized coordinates $\{\hat{b}^n_{k'}\}^{N,K'}_{n=1,k'=1}$ and RGB patches $\{\hat{p}^n_{k'}\}^{N,K}_{n=1,k'=1}$ at all predicted $K'$ frames.

## D  PROGRAM PARSER

Following Yi et al. (2020), we use a seq2seq model (Bahdanau et al., 2015) with attention mechanism to word sequences into a set of symbolic programs and treat questions and choices, separately. The model consists of a Bi-LSTM (Graves et al., 2005) to encode the word sequences into hidden states and a decoder to attentively aggregate the important words to decode the target program. Specifically, to encode the word embeddings $\{w_i\}^I_{i=1}$ into the hidden states, we have

$$\overrightarrow{e}_i, \overrightarrow{h}_i = \overrightarrow{\mathrm{LSTM}}(f^{enc}_w(w_i), \overrightarrow{h}_{i-1}), \quad \overleftarrow{e}_i, \overleftarrow{h}_i = \overleftarrow{\mathrm{LSTM}}(f^{enc}_w(w_i), \overleftarrow{h}_{i+1}), \quad (7)$$

where $I$ is the number of words and $f^{enc}_w$ is an encoder for word embeddings. To decode the encoded vectors $\{e_i\}^I_{i=1}$ into symbolic programs $\{p_j\}^J_{j=1}$, we have

$$q_j = \mathrm{LSTM}(f^{dec}_c(p_{j-1})), \quad \alpha_{i,j} = \frac{exp(q^T_j e_i)}{\sum_i exp(q^T_j e_i)}, \quad \hat{p}_j \sim \mathrm{softmax}(W \cdot (q_j || \sum_i \alpha_{i,j} e_i)), \quad (8)$$

where $e_i = \overrightarrow{e_i} || \overleftarrow{e_i}$ and $J$ is the number of programs. The dimension of the word embedding and all the hidden states is set to 300 and 256, respectively.

## E  CLEVRER OPERATIONS AND PROGRAM EXECUTION

We list all the available data types and operations for CLEVRER VQA (Yi et al., 2020) in Table 8 and Table 7. In this section, we first introduce how we represent the objects, events and moments in the video. Then, we describe how we quantize the static and dynamic concepts and perform temporal and causal reasoning. Finally, we summarize the detailed implementation of all operations in Table 9.

**Representation for Objects, Events and Time.**  We consider a video with $N$ objects and $T$ frames and we sample $K$ frames for collision prediction. The *objects* in Table 8 can be represented by a vector `objects` of length $N$, where `objects`$_n \in [0,1]$ represents the probability of the $n$-h object being referred to. Similarly, we use a vector `events`$^{in}$ of length $N$ to representing the probability of objects coming into the visible scene. We additionally store frame indexes $t^{in}$ for event in, where $t^{in}_n$ indicates the moment when the $n$-th object first appear in the visual scene. We represent event *out* in a similar way as we represent event *in*. For event *collision*, we represent it with a matrix `events`$^{col} \in \mathbb{R}^{N \times N \times K}$, where `events`$^{col}_{n_1,n_2,k}$ represents the $n_1$-th and $n_2$-th objects collide at the $k$-th frame. Since CLEVRER requires temporal relations of *events*, we also maintain a time mask $M \in \mathbb{R}^T$ to annotate valid time steps, where $M_t = 1$ indicates the $t$-th is valid at the current step and $M_t = 0$ indicating invalid. In CLEVRER, `Unique` also involves transformation from *objects* (object set) to *object* (a single object). We achieve by selecting the object with the largest probability. We perform in a similar way to transform *events* to *event*.

**Object and Event Concept Quantization.** We first introduce how DCL quantizes different concepts by showing an example how DCL quantizes the static object concept *cube*. Let $f_n^v$ denote the latent visual feature for the $n$-th object in the video, SA denotes the set of all static attributes. The concept *cube* is represented by a semantic vector $s^{Cube}$ and an indication vector $i^{cube}$. $i^{cube}$ is of length $|SA|$ and L-1 normalized, indicating concept *Cube* belongs to the static attribute Shape. We compute the confidence scores that an object is a *Cube* by

$$P_n^{cube} = \sum_{sa \in SA} (i_{sa}^{Cube} \frac{cos(s^{cube}, m^{sa}(f_n^v)) - \delta}{\lambda}), \tag{9}$$

where $\delta$ and $\lambda$ denotes the shifting and scaling scalars and are set to 0.15 and 0.2, respectively. $cos()$ calculates the cosine similarity between two vectors and $m^{sa}$ denotes a linear transformation, mapping object features into the concept representation space. We get a vector of length $N$ by applying this concept filter to all objects, denoted as $ObjFilter(cube)$.

We perform similar quantization to temporal dynamic concepts. For event *in* and *out*, we simply replace $f_n^v$ with temporal sequence features $f_n^s \in \mathbb{R}^{4T}$ to get $events_n^{in}$. For event *collision*, we replace $f_n^v$ with $f_{n_1, n_2, k}^c$ to predict the confidence that the $n_1$-th and the $n_2$-th objects collide at the $k$-frame and get $events_{n_1, n_2, k}^{out}$. For moment-specific dynamic concepts *moving* and *stationary*, we adopt frame-specific feature $f_{n,t^*}^s \in \mathbb{R}^{4T}$ for concept prediction. We denote the filter result on all objects as $ObjFilter(moving, t^*)$. Specifically, we generate the sequence feature $f_{n,t^*}^s$ at the $t^*$-th frame by only concatenating $[x_t^n, y_t^n, w_t^n, h_t^n]$ from $t^* - \tau$ to $t^* + \tau$ frames and padding other dimensions with $\mathbf{0}$.

**Temporal and causal Reasoning.** One unique feature for CLEVRER is that it requires a model to reason over temporal and causal structures of the video to get the answer. We handle Filter_before and Filter_after by updating the valid time mask $M$. For example, to filter events happening after a target event. We first get the frame $t^*$ that the target event happens at and update valid time mask $M$ by setting $M_t = 1$ if $t > t^*$ else $M_t = 0$. We then ignore the *events* happening at the invalid frames and update the temporal sequence features to be $f_n^{s'} = f_n^s \circ M_{exp}$, where $\circ$ denotes the component-wise multiplication and $M_{exp} = [M; M; M; M] \in \mathbb{R}^{4T}$.

For Filter_order of $events^{type}$, we first filter all the valid events by find events who $event^{type} > \eta$. $\eta$ is simply set to 0 and $type \in \{in, out, collision\}$. We then sort all the remain events based on $t^{type}$ to find the target event.

For Filter_ancestor of a *collision event*, we first predict valid events by finding $events_{type} > \eta$. We then return all valid events that are in the causal graphs of the given *collision* event.

We summarize the implementation of all operations in Table 9.

# F    TRAJECTORY PERFORMANCE EVALUATION.

In this section, we compare different kinds of methods for generating object trajectory proposals. Greedy+IoU denotes the method used in (Gkioxari & Malik, 2015), which adopts a greedy Viterbi algorithm to generate trajectories based on IoUs of image proposals in connective frames. Greedy+IoU+Attr. denotes the method adopts the greedy algorithm to generate trajectory proposals based on the IoUs and predicted static attributes. LSM+IoU denotes the method that we use linear sum assignment to connect the image proposals based on IoUs. LSM+IoU+Attr. denotes the method we use linear sum assignment to connect image proposals based on IoUs and predicted static attributes. LSM+IoU+Attr.+KF denotes the method that we apply additional Kalman filtering (Kalman, 1960; Bewley et al., 2016; Wojke et al., 2017) to LSM+IoU+Attr.. We evaluate the performance of different methods by compute the IoU between the generated trajectory proposals and the ground-truth trajectories. We consider it a "correct" trajectory proposal if the IoU between the proposal and the ground-truth is larger than a threshold. Specifically, two metrics are used evaluation, $precision = \frac{N_{correct}}{N_p}$ and $recall = \frac{N_{correct}}{N_{gt}}$, where $N_{correct}$, $N_p$ and $N_{gt}$ denotes the number of correct proposals, the number of proposals and the number of ground-truth objects, respectively.

Table 10 list the performance of different thresholds. We can see that Greedy+IoU achieve bad performance when the IoU threshold is high while our method based on linear sum assignment and

static attributes are more robust. Empirically, we find that linear sum assignment and static attributes can help distinguish close object proposals and make the correct image proposal assignments. Similar to normal object tracking algorithms (Bewley et al., 2016; Wojke et al., 2017), we also find that adding additional Kalman filter can further slightly improve the trajectory quality.

## G    STATISTICS FOR CLEVRER-GROUNDING AND CLEVRER-RETRIEVAL

We simply use the videos from original CLEVRER training set as the training videos for CLEVRER-Grounding and CLEVRER-Retrieval and evaluate their performance on the validation set. CLEVERER-Grounding contains 10.2 expressions for each video on average. CLEVERER-Retrieval contains 7.4 expressions for each video in the training set. We for evaluating the video retrieval task on the validation set. We evaluate the performance of CLEVERER-Grounding task on all 5,000 videos from the original CLEVRER validation set. For CLEVERER-Retrieval, We additionally generate 1,129 unique expressions from the validation set as query and treat the first 1,000 videos from CLEVRER validation set as the gallery. We provide more examples for CLEVRER-Grounding and CLEVRER-Retrieval datasets in Fig. 5, Fig. 6 and Fig. 7. It can be seen from the examples that the newly proposed CLEVRER-Grounding and CLEVRER-Retrieval datasets contain delicate and compositional expressions for objects and physical events. It can evaluate models' ability to perform compositional temporal and causal reasoning.

## H    TRAINING OBJECTIVES

In this section, we provide the explicit training objectives for each module. We optimize the feature extractor and the concept embeddings in the executors by question answering. We treat each option of a multiple-choice question as an independent boolean question during training and we use different loss functions for different question types. Specifically, we use cross-entropy loss to supervise open-ended questions and use mean square error loss to supervise counting questions. Formally, for open-ended questions, we have

$$\mathcal{L}_{QA,open} = -\sum_{c=1}^{C} \mathbb{1}\{y_a = c\} \log(p_c), \tag{10}$$

where $C$ is the size of the pre-defined answer set, $p_c$ is the probability for the $c$-th answer and $y_a$ is the ground-truth answer label. For counting questions, we have

$$\mathcal{L}_{QA,count} = (y_a - z)^2, \tag{11}$$

where $z$ is the predicted number and $y_a$ is the ground-truth number label.

We train the program parser with program labels using cross-entropy loss,

$$\mathcal{L}_{program} = -\sum_{j=1}^{J} \mathbb{1}\{y_p = j\} \log(p_j), \tag{12}$$

where $J$ is the size of the pre-defined program set, $p_j$ is the probability for the $j$-th program and $y_p$ is the ground-truth program label.

We optimize the dynamic predictor with mean square error loss. Mathematically, we have

$$\mathcal{L}_{dynamic} = \sum_{n=1}^{N} \| b^n - \hat{b}^n \|_2^2 + \sum_{n=1}^{N} \sum_{i_1=1}^{N_p} \sum_{i_2=1}^{N_p} \| p_{i_1,i_2}^n - \hat{p}_{i_1,i_2}^n \|_2^2, \tag{13}$$

where $b^n$ is the object coordinates for the $n$-th object, $p_{i_1,i_2}^n$ is the pixel value of the $n$-th object's cropped patch at $(i_1, i_2)$, and $N_p$ is the cropped size. $\hat{b}^n$ and $\hat{p}_{i_1,i_2}^n$ are the dynamic predictor's predictions for $b^n$ and $p_{i_1,i_2}^n$.

| Type | Operation | Signature |
|---|---|---|
| Input Modules | `Objects`
Returns all objects in the video | $() \rightarrow objects$ |
| | `Events`
Returns all events happening in the video | $() \rightarrow events$ |
| | `UnseenEvents`
Returns all future events happening in the video | $() \rightarrow events$ |
| | `Start`
Returns the special "start" event | $() \rightarrow event$ |
| | `end`
Returns the special "end" event | $() \rightarrow event$ |
| Object Filter Modules | `Filter_static_concept`
Select objects from the input list with the input static concept | $(objects, concept) \rightarrow objects$ |
| | `Filter_dynamic_concept`
Selects objects in the input frame with the dynamic concept | $(objects, concept, frame) \rightarrow objects$ |
| Event Filter Modules | `Filter_in`
Select incoming events of the input objects | $(events, objects) \rightarrow events$ |
| | `Filter_out`
Select existing events of the input objects | $(events, objects) \rightarrow events$ |
| | `Filter_collision`
Select all collisions that involve an of the input objects | $(events, objects) \rightarrow events$ |
| | `Get_col_partner`
Return the collision partner of the input object | $(event, object) \rightarrow object$ |
| | `Filter_before`
Select all events before the target event | $(events, events) \rightarrow events$ |
| | `Filter_after`
Select all events after the target event | $(events, events) \rightarrow events$ |
| | `Filter_order`
Select the event at the specific time order | $(events, order) \rightarrow event$ |
| | `Filter_ancestor`
Select all ancestors of the input event in the causal graph | $(event, events) \rightarrow events$ |
| | `Get_frame`
Return the frame of the input event in the video | $(event) \rightarrow frame$ |
| Output Modules | `Query_Attribute`
Returns the query attribute of the input object | $(object) \rightarrow concept$ |
| | `Count`
Returns the number of the input objects/ events | $(objects) \rightarrow int$
$(events) \rightarrow int$ |
| | `Exist`
Returns "yes" if the input objects is not empty | $(objects) \rightarrow bool$ |
| | `Belong_to`
Returns "yes" if the input event belongs to the input event sets | $(event, events) \rightarrow bool$ |
| | `Negate`
Returns the negation of the input boolean | $(bool) \rightarrow bool$ |
| | `Unique`
Return the only event /object in the input list | $(events) \rightarrow event$
$(objects) \rightarrow object$ |

Table 7: Operations available on CLEVRER dataset.

| Type | Semantics |
|---|---|
| *object* | A single object in the video. |
| *objects* | A set of objects in the video. |
| *event* | A single event in the video. |
| *events* | A set of events in the video. |
| *order* | The chronological order of an event, e.g. "First", "Second" and "Last". |
| *static concept* | Object-level static concepts like "Red", "Sphere" and "Mental". |
| *dynamic concept* | Object-level dynamic concepts like "Moving" and "Stationary". |
| *attribute* | Static attributes including "Color", "Shape" and "Material". |
| *frame* | The frame number of an event. |
| *int* | A single integer like "0" and "1". |
| *bool* | A single boolean value, "True" or "False". |

Table 8: The data type system of CLEVRER-VQA.

| Type | Operation/ Signature | Implementation |
|---|---|---|
| Input Modules | `Objects` 
 $() \rightarrow objects$ 
 `Events` 
 $() \rightarrow events$ 
 `UnseenEvents` 
 $() \rightarrow events$ 
 `Start` 
 $() \rightarrow M$ 
 `end` 
 $() \rightarrow M$ | $objects = \mathbf{1}$ 

 $events^{type}$ for $type \in \{in, out, col.\}$ 

 $events^{col'}$ and $events^{out'}$ 

 $M_t = 1$ if $t < 5$ else $M_t = 0$ 

 $M_t = 1$ if $t > (T-5)$ else $M_t = 0$ |
| Object Filter Modules | `Filter_static_concept` 
 (objs: *objects*, sa: *concept*) $\rightarrow$ *objects* 
 `Filter_dynamic_concept` 
 (objs: *objects*, da: *concept*, t: *frame*) $\rightarrow$ *objects* | $\min(objs, \text{ObjFilter}(sa))$ 

 $\min(objects, \text{ObjFilter}(da,\text{t}))$ |
| Event Filter Modules | `Filter_in` 
 (events$^{in}$ : *events*, objs: *objects*) $\rightarrow$ *events* 
 `Filter_out` 
 (events$^{out}$ : *events*, objs: *objects*) $\rightarrow$ *events* 
 `Filter_collision` 
 (events$^{col}$: *events*, objs: *objects*) $\rightarrow$ *events* 
 `Get_col_partner` 
 (events$^{col}$ : *events*, $obj_n$ : *object*) $\rightarrow$ *objects* 
 `Filter_before` 
 ($events_n^{in}$ : *events*, event1: *event*) 
 `Filter_after` 
 ($events_n^{in}$ : *events*, event1: *event*) $\rightarrow$ *events* 
 `Filter_order` 
 ($events_n^{in}$ : *events*, or: *order*) $\rightarrow$ *event* 
 `Filter_ancestor` 
 (event1: *event*, events1: *events*) $\rightarrow$ *events* 
 `Get_frame` 
 (event1: *event*) $\rightarrow$ *frame* | $\min(objs, \text{events}^{in})$ 

 $\min(objs, \text{events}^{out})$ 

 $\min(objs^{exp}, events^{col.})$ 

 $\max_{k \in [1,K]}(\text{events}_{n,k}^{col})$ 

 $events_n^{in} = -1$ if $t_n^{in} > t^{event1}$ 

 $events_n^{in} = -1$ if $t_n^{in} < t^{event1}$ 

 $events_n^{in} > 0$ if $order_n^{in} = or$ 

 $\{\text{event1}_n > 0$ and $\text{events1}_n$ in the causal graph of event1$\}$ 
 $t^{event1}$ |
| Output Modules | `Query_Attribute` 

 (obj: *object*, a: *attribute*) $\rightarrow$ *concept* 
 `Count` 
 (objs: *objects*) $\rightarrow$ *int* 
 `Exist` 
 (objs: *objects*) $\rightarrow$ *bool* 
 `Belong_to` 
 (event1: *event*, events1: *events*) $\rightarrow$ *bool* 
 `Negate` 
 (bl: *bool*) $\rightarrow$ *bool* | $P^{op} = \frac{ObjFilter(op) \cdot i_a^{op}}{\sum_{op'} ObjFilter(op') \cdot i_a^{op'}}$ 

 $\sum_n(\text{objs}_n > 0)$ 

 $(\sum_n(\text{objs}_n > 0)) > 0$ 

 True if event1 $\in$ events1 else False 

 False if bl else True |

Table 9: Neural operations in DCL. $events^{col'}$ denotes the *collision* events happening at the unseen future frames. $objs^{exp} \in \mathbb{R}^{N \times N \times K}$ and $objs_{n_1,n_2,k}^{exp} = \max(objs_{n_1}, objs_{n_2})$. $events_{n,k}^{col}$ denotes all the collision events that the $n$-th object get involved at the $k$-th frame.

| | 0.5 | | 0.6 | | 0.7 | | 0.8 | | 0.9 | |
|---|---|---|---|---|---|---|---|---|---|---|
| | prec. | recall | prec. | recall | prec. | recall | prec. | recall | prec. | recall |
| Greedy+IoU | 87.2 | 88.3 | 71.4 | 72.3 | 57.3 | 58.0 | 46.9 | 47.5 | 39.6 | 40.0 |
| Greedy+IoU+Attr. | 97.0 | 97.4 | 95.1 | 95.5 | 93.1 | 93.5 | 89.9 | 90.3 | 83.6 | 83.9 |
| LSM+IoU | 97.6 | **98.8** | 96.9 | **98.1** | 96.0 | 97.2 | 93.8 | 95.0 | 88.5 | 89.6 |
| LSM+IoU+Attr. | **99.1** | 98.4 | **98.6** | 97.9 | 97.9 | 97.2 | **96.2** | 95.5 | 91.5 | 90.8 |
| LSM+IoU+Attr.+KF | **99.1** | 98.4 | **98.6** | 97.9 | **98.0** | **97.3** | **96.2** | **95.6** | **91.6** | **90.9** |

Table 10: The evaluation of different methods for object trajectory generation.

**Query:** *The collision that happens before the gray object enters the scene.*

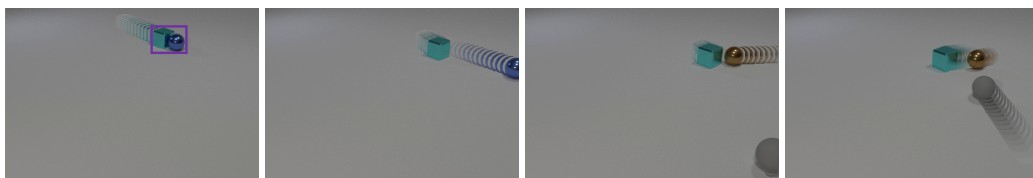

**Query:** *The green object enters the scene before the rubber sphere enters the scene*

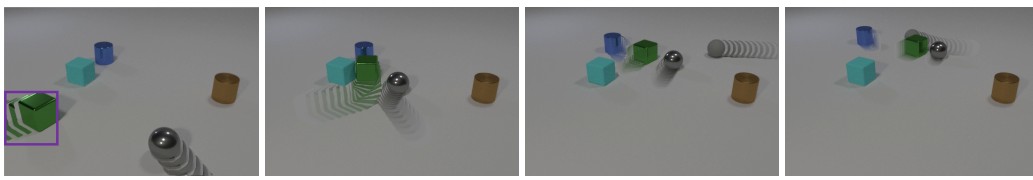

**Query:** *The cube exits the scene after the sphere enters the scene*

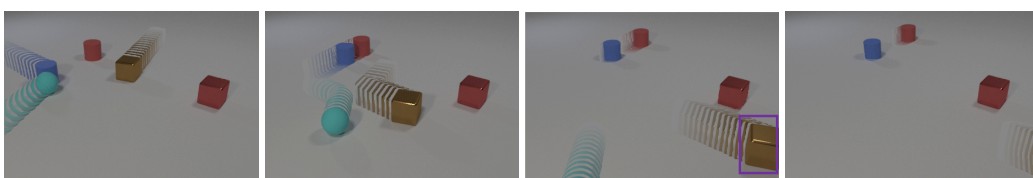

**Query:** *The metal cylinder that is stationary when the sphere enters the scene*

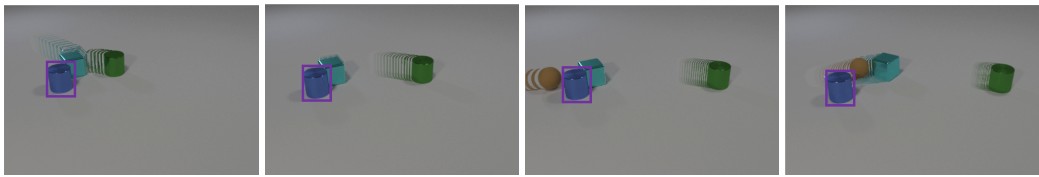

Figure 5: Typical examples of CLEVRER-Grounding datasets. The target regions are bounded with purple boxes.

**Query:** *A video that contains a collision that happens after the yellow metal cylinder enters the scene.*

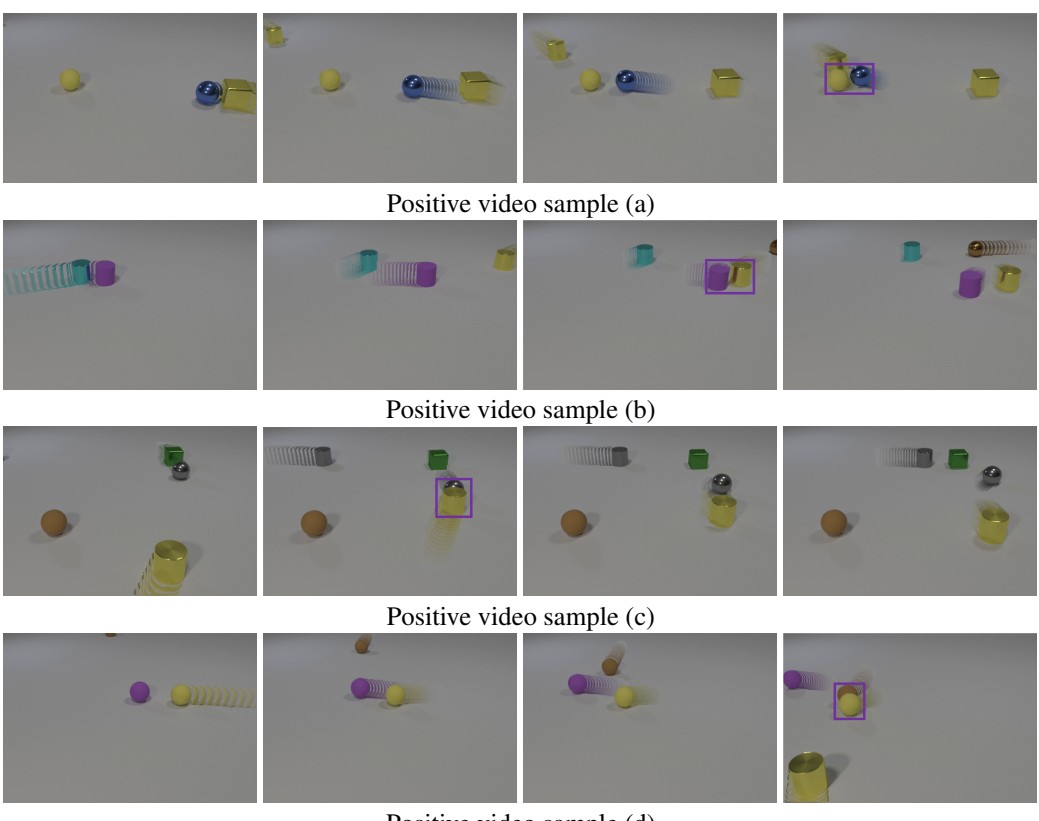

Positive video sample (a)

Positive video sample (b)

Positive video sample (c)

Positive video sample (d)

Figure 6: A exemplar query expression and 4 of its associated positive videos from CLEVRER-Retrieval dataset. The target regions in videos are bounded with purple boxes.

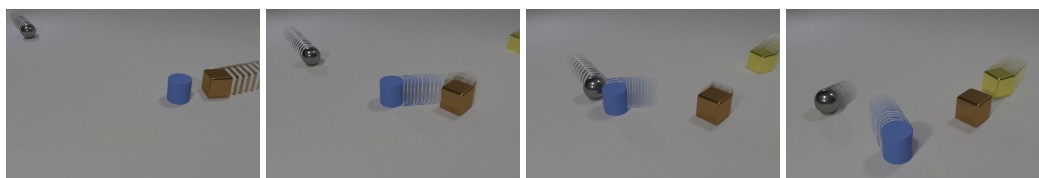

1. *A video that contains an object that collides with the brown metal cube.*
2. *A video that contains an object that collides with the gray metal sphere.*
3. *A video that contains an object to collide with the brown metal cube.*
4. *A video that contains an object to collide with the gray metal sphere.*
5. *A video that contains a collision that happens after the yellow metal cube enters the scene.*
6. *A video that contains a collision that happens after the brown metal cube enters the scene.*
7. *A video that contains a collision that happens before the yellow metal cube enters the scene.*

Figure 7: An typical query video and its associated positive expressions from CLEVRER-Retrieval dataset.

