# OpenReview forum: "Grounding Physical Concepts of Objects and Events Through Dynamic Visual Reasoning"
_ICLR.cc/2021/Conference — ICLR 2021 Poster_

### Official Review · AnonReviewer1 · 2020-10-28
**I find the method is interesting but the paper could be improved on its current form**

**Rating:** 6
**Confidence:** 4

**Review:**

Summary:

The paper presents a neural-symbolic approach for video question answering. In particular, the authors propose the Dynamic Concept Learner (DCL), a network architecture to localize object trajectories over time and further build a graph network to model the physical interaction between objects and events. Finally, a standard symbolic executor is used to reason across visual and language cues for question answering. The proposed method is evaluated on CLEVRER dataset in which it achieves new state of the art performance.

Pros:

(1) The main contribution of this paper is that it is able to ground object and event concepts to model the dynamic scene of the interactions between objects/events without the use of extra supervision. This is well motivated as we do not always have access to explicit labels such as object attributes, masks, and collision events in real-world applications.

(2) The idea of dynamics prediction to support predictive and counterfactual questions is new. This makes the proposed framework as a whole interesting IMHO.

(3) The proposed method sets new state of the art performance on CLEVRER dataset in all question types. On video grounding and video-text retrieval tasks, DCL seems to achieve interesting results.

Cons:

Despite the pros above, I have some comments on the paper:

(1) I found the paper is quite hard to follow. Figure 2 and its description represent the process of DCL (object trajectory detector -> dynamic predictor -> feature extractor -> symbolic executor). However, the multi-step training paradigm in Sec. 1, 3.1, and 3.2 seems to follow a different structure. In addition, some details are missing in the paper. For example, in the ‘Grounding Object and Event Concepts’ subsection, where did you get the vector embedding for the concept "moving" (s_moving) from?

(2) The motivation in the subsection ‘Trajectory refinement’ is exactly as what explained in DeepSORT [1] – a common tracking algorithm. In addition, I believe the use of Kalman Filters in DeepSORT makes it better in handling occlusion than the Eq. 2. I suggest the author discuss the object tracking literature for better judgments.

(3) One of the main weaknesses of this paper is they only evaluate their method on one synthetic dataset which is CLEVRER. It seems to me that the proposed method is particularly designed to solve this dataset. While I find the results interesting, I think physical interactions in those synthetic data are too simple and do not reflect much what happens in real-world data. Not to mention the limit in terms of linguistic variants of those synthetic datasets compared to human natural language. I would appreciate if the author can evaluate the proposed model on other datasets such as CATER [2] for temporal reasoning and Video QA datasets (TGIF-QA [3], TVQA [4], etc).

(4) Similar to my concerns in (3), it would be more convincing to see if the DCL works on video grounding datasets such as ActivityNet-Entities or YouCook2, MSR-VTT for video-text retrieval to support their claims on the generalization of the method.

I am willing to raise the rating if the author could properly address my concerns.

References:
- [1] Wojke, Nicolai, Alex Bewley, and Dietrich Paulus. "Simple online and realtime tracking with a deep association metric." 2017 IEEE international conference on image processing (ICIP). IEEE, 2017.
- [2] Girdhar, Rohit, and Deva Ramanan. "Cater: A diagnostic dataset for compositional actions and temporal reasoning." arXiv preprint arXiv:1910.04744 (2019).
- [3] Jang, Yunseok, et al. "Tgif-qa: Toward spatio-temporal reasoning in visual question answering." Proceedings of the IEEE Conference on Computer Vision and Pattern Recognition. 2017.
- [4] Lei, J., Yu, L., Bansal, M., & Berg, T. L. (2018). Tvqa: Localized, compositional video question answering. arXiv preprint arXiv:1809.01696.

---

> ### Author Response · Authors · 2020-11-20
> **Response to AnonReviewer1**
>
> Thanks for your interest in the method and your detailed and constructive comments.
>
>
> > The details of the concept embedding.
>
> We store a vector embedding for each concept in the symbolic executor.
> During program execution, we quantize concepts by matching the concept embeddings like "moving" with object features.
> DCL optimizes the concept embeddings by question answering.
>
> >Paper organization and the details of DCL.
>
> The captions of Fig. 2 shows the inference process for a counterfactual question (object trajectory detector -> dynamic predictor -> feature extractor -> symbolic executor).
>
> During training, we follow a multi-step training strategy,
> - Step 1: we learn the static concept embeddings (color, shape, and material) in the symbolic executor by descriptive and explanatory question-answer pairs. It requires no dynamic prediction (coarse trajectory detector in Eq. (1) -> feature extractor -> symbolic executor).
> - Step 2: we then use the learned concept embeddings to quantize static concepts for each object and refine the object trajectories (see Eq. (2)-(3)).
> - Step 3: we train a dynamic predictor based on the refined trajectories.
> - Step 4: we train the full model with all the question-answer pairs(object trajectory detector -> dynamic predictor ->feature extractor -> symbolic executor).
>
> We will adjust the caption in Fig. 2 and provide more details in Sec. 3.1 and Sec. 3.2 accordingly in the revised version.
>
> >The motivation of Trajectory refinement.
>
> Thanks for pointing out the object tracking literature (DeepSORT).
> We will discuss and cite this paper in the revised version.
>
> We would like to clarify that the contribution of "trajectory refinement" in this paper is not the definition of the scoring function in Eq. (2) or the linear sum assignment algorithm for optimization. Our main contribution is to build a framework that can ground static concepts from these coarse object trajectories using questions and answers pairs. These learned concepts can then help refine the object trajectories. We verify the effectiveness of the learned static concepts for better object trajectories in Table 8. It can constantly boost the performance of different optimization solvers.
>
> We will add Kalman Filters into the tracker and analyze its performance in the revised version.
>
> > About the generalization to other real datasets.
>
> We agree that evaluating our model in real-world videos is very important.  We will include new experimental results of DCL on a real-world block-tower video dataset[1]. We believe these results could further verify the effectiveness and generalization of DCL to learn the new dynamic concept in real videos.
>
> > About experiments on other video action datasets.
>
> Please first refer to common concerns to see the importance of learning physical concepts and their differences from human-centric action datasets.
>
> - In Section 4, we have clearly indicated why  CATER is unsuitable for evaluating learning object and event concepts from language.   CATER  is indeed a good dataset to study dynamic and video reasoning, but unfortunately, they do not provide questions and answer pairs.
>
> - We would like to reiterate that our work aims to learn physical objects and event concepts in videos. This is a different research problem from understanding actions and activities. Compared with those action videos, the datasets and the tasks we used in this paper have richer compositional and causal structures.  These are also the main reason why models working well on these action datasets do not achieve good performances on CLEVRER.
>
> [1]. Adam Lerer, Sam Gross, and Rob Fergus. Learning physical intuition of block towers by example. ICML, 2016.
>
> Please let us know if you have additional questions to discuss.

---

> > ### Author Response · Authors · 2020-11-23
> > **Look foward to your response!**
> >
> > Dear Reviewer 1,
> >
> > Thanks again for your constructive review, which has helped us improved the quality and clarity of the paper. In addition to our response above, in the revision, we have added new experiments on a real-world physical dataset and also included comparisons with additional baselines.  The visualization of our model's results could be found on the website (https://dynamicconceptlearner.github.io).
> >
> > As the discussion period is about to end, please don’t hesitate to let us know if there are any additional clarifications that we can offer, as we would love to convince you of the merits of the paper. We appreciate your suggestions. Thanks!

---

> ### Author Response · Authors · 2020-11-23
> **Follow up: Revision Updated.**
>
> Thanks again for your constructive comments. We have made substantial changes in the revision according to your review. Specifically, we have the following changes according to your comments,
>
> - We have added new experiments on a real-world physical dataset [1] to further show DCL’s effectiveness and generalization capacity in (Sec. 4.5, Table 5 and Table 6).
> - We have added a new tracker using Kalman Filtering and analyzed its performance in Appendix F and Table 9.
> - We have provided qualitative visual examples and failure cases for concept learning in a new website (https://dynamicconceptlearner.github.io) to show its interpretability and effectiveness.
> - We have revised the paper carefully, including providing more training and inference details (Sec. 3.2 and appendix H), analysis of the differences between action videos and physical videos (Sec. 2), model limitations (Sec. 5), content adjustment (Sec. 4.5), and grammar checking.
>
> As the discussion period is about to end, please don’t hesitate to let us know if there are any additional clarifications that we can offer, as we would love to convince you of the merits of the paper. Thanks!
>
> [1] Adam Lerer, Sam Gross, and Rob Fergus. Learning physical intuition of block towers by example. ICML, 2016.

---

### Official Review · AnonReviewer3 · 2020-10-28
**An effective approach to learning dynamic concepts without expensive supervision**

**Rating:** 7
**Confidence:** 3

**Review:**

=Summary
The paper proposes a new framework, Dynamic Concept Learner (DCL), which learns by watching videos and reading questions/answers.  It is inspired by prior work which combines symbolic representations with video dynamics modeling, but unlike prior work, here the authors do not use any additional supervision except for question/answer pairs. DCL is quiet complex, involving multiple components, including program parser (transforms the question/answer into executable symbolic programs), object detection&tracking, object-centric feature extractor (with static and dynamic components), object/event concept embeddings (share the same latent space as the visual features), trajectory refinement (based on static cues) and dynamic predictor (for future and counterfactual scenes), and, finally, symbolic executor (to predict whether the answer is correct), which runs on the latent features. DCL achieves state-of-the-art results on the CLEVRER dataset. The learned representations are also shown to be useful for other tasks, such as grounding and retrieval.

=Strengths

The paper is overall well written and the approach is novel to the best of my knowledge.

DCL achieves state-of-the-art results on the CLEVRER dataset, w/o accessing visual labels from simulation during training.
The biggest improvement is notably achieved on the predictive and counterfactual questions.

The authors also show that DCL learns to ground object/event concepts w/o explicit labels.

Finally, the authors construct CLEVRER-Grounding and CLEVRER-Retrieval datasets to test the generalization ability of DCL. They show that it significantly improves over a baseline on both tasks.

=Weaknesses/High-level comments

Counterfactual questions remain particularly hard to answer, do the authors have some intuition as to what may be missing in their model? What are some common failure modes?

Table 1: it is somewhat surprising that the difference between the DCL and DCL-Oracle is so small, and DCL even improves over DCL-Oracle in one case. Why is that?

What do the authors believe is the main limitation of their proposed approach? Currently it operates on synthetic images as well as synthetic language, what will be necessary to move to more realistic domains?

Nothing was stated about making the code available.

=Minor comments (P# - page number)
- P1 that what the concepts => what the concepts
- P3, Sec 3: DYNAMICS CONCEPT LEARNER => DYNAMIC CONCEPT LEARNER
- P4: an program parser => a program parser
- The paper heavily relies on the supplemental material.

---

> ### Author Response · Authors · 2020-11-21
> **Response to AnonReviewer3**
>
> Thanks for your sincere and constructive comments.
>
> > About the limitation of the current DCL model, performance on counterfactual questions, and common failure cases.
>
> To achieve high QA accuracy in CLEVRER, a model mainly requires two kinds of abilities: 1). physical object and event concept learning, and 2). physical dynamic prediction.
> While our DCL can effectively ground object and event concepts (see Table 2), it still requires a stronger ability for long-term video dynamic prediction to achieve higher accuracy on counterfactual questions.
> Note that long-term dynamic video prediction [1,2] has always been a difficult research topic.
>
> We will show some typical failure cases for counterfactual questions in the revised version.
>
> [1]. Li, Yunzhu, et al. "Propagation networks for model-based control under
> partial observation." ICRA, 2019.
>
> [2]. Qi, Haozhi, et al. "Learning Long-term Visual Dynamics with Region Proposal Interaction Networks." arXiv preprint, 2020.
>
> > About the small performance difference between DCL and DCL-Oracle.
>
> Compared with DCL, DCL-Oracle uses additional supervision for physical object and event concepts during training.
> DCL can effectively ground physical object and event concepts through only question answering and achieve high concept recognition accuracy (see Table 2). Thus, the bottleneck for the QA accuracy is not the DCL's ability to ground physical concepts but to learn physical dynamics.
> Since both DCL and DCL-Oracle use PropNet[1] as the dynamic prediction model, the performance gap is small.
>
> [1]. Li, Yunzhu, et al. "Propagation networks for model-based control under
> partial observation." ICRA, 2019.
>
> > About more experiments on real-world videos
>
> We will conduct new experiments on real videos[1] to show DCL’s effectiveness and generalization.
>
> [1]. Adam Lerer, Sam Gross, and Rob Fergus. Learning physical intuition of block towers by example. ICML, 2016.
>
> > About code and data.
>
> We will make our code, models and the new datasets available as soon as possible.
>
> > About typos and reliance on the supplemental material.
>
> Thanks for the advice on writing. We will fix the grammar typos and make use of the additional page during the rebuttal for a better revised paper.

---

> ### Author Response · Authors · 2020-11-24
> **[Revision Updated] look forward to your feedback!**
>
> Dear Reviewer 3,
>
> Thanks again for your detailed and positive comments.
> We have made substantial changes in the revision according to your review.
> In particular, we have the following changes according to your comments,
>
> - We have added new experiments on a real-world physical dataset [1] to further show DCL’s effectiveness and generalization capacity in (Sec. 4.5, Table 5 and Table 6).
> - We have revised the paper carefully, including providing more training and inference details (Sec. 3.2 and appendix H), analysis of the differences between action videos and physical videos (Sec. 2) and model limitations (Sec. 5).
> - We have provided qualitative visual examples and failure cases for concept learning in a new website (https://dynamicconceptlearner.github.io) to show its interpretability and effectiveness.
> - We will release our code, data and models as soon as possible.
>
> As the discussion period is about to end, please don’t hesitate to let us know if there are any additional clarifications that we can offer, as we would love to convince you of the merits of the paper. Thanks!
>
> [1] Adam Lerer, Sam Gross, and Rob Fergus. Learning physical intuition of block towers by example. ICML, 2016.

---

### Official Review · AnonReviewer2 · 2020-10-29
**Good paper on dynamic visual reasoning but with concerns in generalization to real-world videos**

**Rating:** 7
**Confidence:** 4

**Review:**

> Summary:

The paper studies the problem of dynamic visual reasoning on the recently proposed synthetic video QA dataset CLEVRER for understanding visual properties, physical events, the causal relationship between events, and making future and counterfactual predictions. The paper proposes a joint framework called Dynamic Concept Learner which contains five modules (object trajectory detector, video feature extractor, dynamic predictor, language program parser, and neural symbolic executor) and follows a multi-step training paradigm to train the model purely from the question-answer pairs in CLEVRER train split.  The object trajectory detector module, the concept embedding/quantization, and the dynamic predictor module play a very important role in DCL. These modules enable the dynamic & causal reasoning abilities for DCL. The neural program parsing & execution part of DCL is similar to that in Neuro-Symbolic Concept Learner (NS-CL) (Mao et al., 2019) and thus DCL can be regarded as the dynamic extended version of  NS-CL.

> Strengths:

*  The paper is well-written and easy to follow.  A lot of details about the DCL model, neural programs, and dataset are provided in the appendix.

* The experiment on CLEVRER shows the effectiveness of DCL over previous methods.  Additional experiments on new CLEVRER Grounding & Retrieval applications demonstrate the generalization ability of DCL trained with the original VQA task on other CLEVRER related tasks.

* DCL is trained directly from QA pairs without additional labels for object attributes and events but still outperforms the previous methods, which is a big strength.

* During inference, DCL can acquire attribute/event concept perception, object trajectory prediction at the same time. Once the answer is generated, the whole reasoning path for the answer is also ready, which exhibits the good model interpretability of DCL.

* From *VirTex: Learning Visual Representations from Textual Annotations (Desai et al., 2020)* and *LXMERT: Learning Cross-Modality Encoder Representations from Transformers (Tan et al., 2020)*, we can see that actually textual annotations such captions and question-answer pairs contain high-quality information and are helpful for learning visual or concept representations. I think this paper is another good example of discovering such high-quality information without using additional fine-grained labels.

> Some concerns and suggestions:

- My major concern is that the evaluation is based on a synthetic toy video dataset and it is simplistic compared to real-world videos. Although the author demonstrates that DCL has transferability between different vision-language tasks on CLEVRER, it is still skeptical whether DCL can be generalized to real-world video QA datasets with more complex visual scenes such as TGIF-QA, MSVD-QA, or MSRVTT-QA since the dynamic attributes and events in CLEVRER is only limited to (moving, stationary) and (in, out, collision). If possible, evaluating on more datasets would further demonstrate the effectiveness of DCL across different visual scenes.

* Besides NS-DR baseline which was proposed together with CLEVRER dataset, Memory (Fan et al., 2019) is the most recent visual-language reasoning approach for videos. It is evaluated on TGIF-QA, MSVD-QA, and  MSRVTT-QA in the original paper. I think the author should find some newer approaches evaluated on these datasets to make comparisons on CLEVRER. For instance, [Hierarchical Conditional Relation Networks for Video Question Answering (CVPR 20’)](https://openaccess.thecvf.com/content_CVPR_2020/papers/Le_Hierarchical_Conditional_Relation_Networks_for_Video_Question_Answering_CVPR_2020_paper.pdf) can be a candidate baseline.

* Although the author introduces all the five modules and describes the training steps, it is still not clear for readers to understand how to train the five modules jointly without providing an explicit loss function or training objective. I think the author should specify such a training objective explicitly.

- The paper states that DCL does not require additional fine-grained labels for attributes and events. However, DCL requires additional 1000 programs to train the program parser for all question types. Can we call these 1000 programs additional annotations? For a new visual scene that is different from the current 3D synthetic domain, to make DCL suitable for the new visual domain,  we need to define new sets of programs and concepts, which may require additional annotation labors.

- To make the experiments more convincing on Retrieval and Grounding tasks, the author could further compare with some recent video-text retrieval/grounding approaches. To name a few: 1) [Fine-grained Video-Text Retrieval with Hierarchical Graph Reasoning](https://openaccess.thecvf.com/content_CVPR_2020/papers/Chen_Fine-Grained_Video-Text_Retrieval_With_Hierarchical_Graph_Reasoning_CVPR_2020_paper.pdf) 2) [TVR: A Large-Scale Dataset for Video-Subtitle Moment Retrieval](https://arxiv.org/pdf/2001.09099.pdf) 3) [Semantic Conditioned Dynamic Modulation for Temporal Sentence Grounding in Videos](https://papers.nips.cc/paper/8344-semantic-conditioned-dynamic-modulation-for-temporal-sentence-grounding-in-videos.pdf)

---

> ### Author Response · Authors · 2020-11-21
> **Response to AnonReviewer2**
>
> Thanks for your detailed and constructive comments.
>
> > About experiments on real-world datasets.
>
> We agree that evaluating DCL on more datasets would further demonstrate its effectiveness across different visual scenes.
> We will conduct experiments on a real physical dataset[1] to show the model's effectiveness and generalization.
>
> [1]. Adam Lerer, Sam Gross, and Rob Fergus. Learning physical intuition of block towers by example. ICML, 2016.
>
> > About more recent baselines on CLEVRER-QA.
>
> We will apply a new recent VQA baseline HCRN to the CLEVRER and analyze its performance.
> We will cite and discuss the new baseline HCRN[1] in the revised version.
>
> [1]. Le, Thao Minh, et al. "Hierarchical Conditional Relation Networks for Video Question Answering." CVPR, 2020.
>
> > About the explicit loss functions.
>
> We will provide explicit loss functions for each module in the revised version.
>
> > About the requirement of annotation for program parser training.
>
> Yes, our DCL requires 1,000 annotated programs for training the program parser.
> It should also be noted that some previous research like [1] has been learning to parse questions into symbolic programs by only question answering.
> They achieved this goal with REINFORCE[2] and curriculum learning.
> We regard learning the program parser by question answering for CLEVRER as an interesting future direction.
>
> [1]. Mao, Jiayuan, et al. "The neuro-symbolic concept learner: Interpreting scenes, words, and sentences from natural supervision." ICLR, 2019.
>
> [2]. Ronald J Williams. Simple statistical gradient-following algorithms for connectionist reinforcement learning. MLJ, 1992.
>
>
> > About the new baseline for video-text retrieval.
>
> We will implement HGR[1] for CLEVRER-Retrieval, cite and analyze its performance in the revised version.
>
> [1]. Chen, Shizhe, et al. "Fine-grained Video-Text Retrieval with Hierarchical Graph Reasoning." CVPR, 2020.

---

> ### Author Response · Authors · 2020-11-24
> **[revision updated] look forward to your feedback**
>
> Dear Reviewer 2,
>
> Thanks again for your detailed and constructive comments.
> We have made substantial changes in the revision according to your review.
> In particular, we have the following changes according to your comments,
>
> - We have added new experiments on a real-world physical dataset [1] to further show DCL’s effectiveness and generalization capacity in (Sec. 4.5, Table 5 and Table 6).
> - We have implemented some more recent baseline methods [2,3] to CLEVRER-QA (Sec. 4.2 and Table 1) and CLEVRER-Retrieval (Sec. 4.4 and Table 4) and analyze their performance.
> - We have revised the paper carefully, including providing explicit training loss functions (Sec. 3.2 and appendix H), analysis of the differences between action videos and physical videos (Sec. 2) and model limitations (Sec. 5).
> - We have provided qualitative visual examples and failure cases for concept learning in a new website (https://dynamicconceptlearner.github.io) to show its interpretability and effectiveness.
>
> As the discussion period is about to end, please don’t hesitate to let us know if there are any additional clarifications that we can offer, as we would love to convince you of the merits of the paper. Thanks!
>
> [1] Adam Lerer, Sam Gross, and Rob Fergus. Learning physical intuition of block towers by example. ICML, 2016.

---

### Official Review · AnonReviewer4 · 2020-10-30
**In this paper, the authors study the problem of dynamic visual reasoning on raw videos. Compared with the previous work which require dense supervision of the video objects and events. Instead they proposed DCL which grounds objects and events without ground truth labels.  The proposed method achieves state of the art performance on CLEVRER and other tasks including video-retrieval and event localization.**

**Rating:** 6
**Confidence:** 2

**Review:**

Pros
1.	This paper extends the semantic modular network on images to the video setting.
2.	By making use of the weak supervision from the question answer, it can learn different kinds of concepts including physical objects, attributes and events.
3.	The paper conduct experiments on different evaluation settings including video causal reasoning, video grounding and generalization experiments.
Cons
1.	The paper lacks some details on the training procedure.
2.	It’s better to show the learned concepts visually as one of the strengthens of is explain ability.
3.	How’s the model’s sample complexity

Comments
1.	Is the trajectory refinement and model learning an iterative process?
2.	What does the RGB patches mean and how do you combine the RGB patches into the predicted locations in the dynamic predictions?
3.	Is this paper using some pre-trained object recognizer and attribute classifier which acts as some kind of pseudo labeling?
4.	In the generalization experiments, does the model trained and test using the same domain dataset?

---

> ### Author Response · Authors · 2020-11-21
> **Response to AnonReviewer4**
>
> Thank you very much for your constructive comments and suggestions.
>
> > About the iterative process for trajectory refinement and model learning.
>
> We adopt a multi-step training paradigm to refine the object trajectories and model learning.
> - Step 1: we simply associate frame object proposals based on object locations  (see Eq. (1)) and get coarse object trajectories.
> - Step 2: we learn static concepts based on the coarse object trajectories and descriptive and explanatory question-answer pairs.
> - Step 3: we use the learned static concepts to refine the object trajectories (see Eq. (2) and Eq. (3)).
> - Step 4: we optimize the full model based on the refined trajectories and all the question-answer pairs.
>
> Theoretically, Step 2 -> Step 3 can be optimized in an iterative fashion. Practically, we only use one-round iteration since we find it sufficient enough to learn the static concepts and the trajectory quality stops increasing.
>
> > About the RGB patches and their combination with the locations in dynamic predictions.
>
> To generate the predictive scenes, we first predict the RGB patches and the location of each object with a Propagation Net.
> We then paste RGB patches back into the predictive locations of the background image for future and counterfactual predictions.
>
> An RGB patch $R^{3 \times N_p \times N_p}$ is a cropped patch centering at the centroid of the target object, where $N\_p$ is the cropped size. We concatenate the RGB patches with the location coordinates by expanding the location coordinates to each pixel of the RGB patch and get a feature of shape $R^{7 \times N_p \times N_p}$ for the Propagation Net. Such a concatenation is also adopted in the previous work NS-DR[1]. We will provide more details on the implementation of Propagation in the revised version.
>
> [1] Kexin Yi, Chuang Gan, Yunzhu Li, Pushmeet Kohli, Jiajun Wu, Antonio Torralba, and Joshua B Tenenbaum. CLEVRER: Collision events for video representation and reasoning. ICLR, 2020.
>
> > About pre-trained object recognizer and attribute classier for pseudo labeling.
>
> Our DCL model doesn't require any object recognizer or attribute classier during training.
> DCL learns to ground and quantize the object attributes by watching videos and answering questions.
> DCL-Oracle shows DCL's performance by adding additional object and event labels.
> We estimate the performance of concept quantization in Table 2 by comparing the predicted concepts with the concept labels provided by the official annotation.
>
> > About the generalization experiments.
>
> In the generalization experiments, the models are trained and tested using the same domain dataset, i.e., the videos in the original CLEVRER-QA dataset and new expressions for grounding and retrieval.
> We claim that DCL has a strong generalization capacity since the learned physical object and event concepts can be applied to new applications, including video retrieval and event localization.
> As stated in the common concerns, we will add new experiments on real videos and a new concept to further show DCL's effectiveness and generalization in the revised version.
> The data samples using in our model and baselines are the same. In practice, we find that our model enjoys better data efficiency than MAC.
>
> > About “to show the learned concepts visually as one of the strengths of is explainability”.
>
> We will provide qualitative visual examples and failure cases for concept learning in a new website to show its interpretability and effectiveness.

---

> ### Author Response · Authors · 2020-11-24
> **[Revision Updated] look forward to your feedback!**
>
> Dear Reviewer 4,
>
> Thanks again for your constructive comments.  We have made substantial changes in the revision according to your review. In particular, we have the following changes according to your comments. 1). We have added new experiments on a real-world physical dataset [1] to further show DCL’s effectiveness and generalization capacity in (Sec. 4.5, Table 5 and Table 6). 2). We have provided qualitative visual examples and failure cases for concept learning in a new website (https://dynamicconceptlearner.github.io) to show its interpretability and effectiveness. 3. We have revised the paper carefully, including providing more training and inference details (Sec. 3.2 and appendix H), analysis of the differences between action videos and physical videos (Sec. 2), and grammar checking.
>
> As the discussion period is about to end, please don’t hesitate to let us know if there are any additional clarifications that we can offer, as we would love to convince you of the merits of the paper. Thanks!
>
> [1] Adam Lerer, Sam Gross, and Rob Fergus. Learning physical intuition of block towers by example. ICML, 2016.

---

### Author Response · Authors · 2020-11-20
**[Pre-revision] General Response to all Reviewers**

We thank all reviewers for their constructive comments.  In addition to the specific response below, we summarize our goals, address some common concerns, and describe the changes to be included in the revision.

### Our Goal

We propose a unified framework to learn physical object and event concepts through dynamic visual reasoning on videos. With these learned concepts, our model can be applied to various problems, including VQA (Sec. 4.2), video grounding (Sec. 4.4), and video retrieval (Sec. 4.4).

### Common Concerns

> Experiments on real videos and new concepts.

Reviewers have raised concerns about DCL’s performance on real videos. We agree that adding new experiments on real-world videos can further prove DCL’s effectiveness and generalization. We are planning to add experiments on a real-world physical video dataset [1] to learn physical dynamic concepts on real videos.

> Differences between physical event understanding, and understanding activities or other video understanding problems

We would like to clarify that the primary goal of our work is to design a unified framework for learning physical object and event concepts (e.g., collision, falling, stability) from videos. These tasks are of great importance in practical applications such as industrial robot manipulation, which requires AI systems with human-like physical common sense. Datasets like ActivityNet, YouCook2, and MSVD-QA focus on other problems of video understanding, such as classifying human actions (e.g., picking up a glass, opening a door), and understanding activities that one or more people engage in over extended periods of time (e.g., running, cooking, cleaning). We agree that understanding actions and activities in the video are important, fundamental research problems in computer vision, but these are not the same research questions we aim to study in this work.  We do believe that the methods we develop should be useful for more general video understanding, but that remains future work.  We also think further work will be needed to build models of intentional action for human-activity event understanding, to complement the physics models we build here for physical event understanding.

### Planned Changes

- We will add new experiments on a real-world physical dataset [1] to further show DCL’s effectiveness and generalization capacity.
- We will implement some more recent baseline methods [2,3] to CLEVRER-QA and CLEVRER-Retrieval and analyze their performance.
- We will add a new tracker using Kalman Filtering and analyze its performance.
- We will revise the paper carefully, including providing more training and inference details, analysis of the differences between - action videos and physical videos, sample complexity, model limitations, content adjustment, and grammar checking.
- We will provide qualitative visual examples and failure cases for concept learning in a new website to show its interpretability and effectiveness.

[1] Adam Lerer, Sam Gross, and Rob Fergus.  Learning physical intuition of block towers by example.  ICML, 2016.

[2] Thao Minh Le and others. Hierarchical Conditional Relation Networks for Video Question Answering. CVPR, 2020.

[3] Shizhe Chen and others. Fine-grained Video-Text Retrieval with Hierarchical Graph Reasoning. CVPR, 2020.

Please don’t hesitate to let us know of any additional comments on the paper or on the planned changes.

---

### Author Response · Authors · 2020-11-21
**Pre-revision Individual Response Updated**

Thank you for your detailed feedback. We have updated the individual response to each of your reviews under your thread. We will update the revision soon. Please let us know if you have further questions.

---

### Author Response · Authors · 2020-11-23
**General Response: Revision Updated**

We would like to thank the reviewers for their thoughtful feedback. We are glad to see that reviewers generally appreciated our paper's contributions and provide constructive and detailed comments for a better revision.

We would like to emphasize again that our primary goal in this paper is to design a unified framework for learning physical object and event concepts (e.g., collision, falling, stability) from videos. These tasks are of great importance in practical applications such as industrial robot manipulation, which requires AI systems with human-like physical common sense.

We have revised our manuscript to include the following changes:

- We have added new experiments on a real-world physical dataset [1] to further show DCL’s effectiveness and generalization capacity in Sec. 4.5. (R1, R2, R3, R4).
- We have implemented some more recent baseline methods [2,3] to CLEVRER-QA (Sec. 4.2 and Table 1)  and CLEVRER-Retrieval (Sec. 4.4 and Table 4)  and analyze their performance. (R2).
- We have added a new tracker using Kalman Filtering and analyzed its performance in Appendix F and Table 9. (R1)
- We have revised the paper carefully, including providing more training and inference details (Sec. 3.2 and appendix H), analysis of the differences between action videos and physical videos (Sec. 2), model limitations (Sec. 5), content adjustment (Sec. 4.5), and grammar checking. (R1, R2, R3, R4)
- We have provided qualitative visual examples and failure cases for concept learning in a new website (https://dynamicconceptlearner.github.io) to show its interpretability and effectiveness. (R1, R2, R3, R4)

Please don’t hesitate to let us know of any additional comments on the manuscript or the changes.


[1] Adam Lerer, Sam Gross, and Rob Fergus. Learning physical intuition of block towers by example. ICML, 2016.

[2] Thao Minh Le and others. Hierarchical Conditional Relation Networks for Video Question Answering. CVPR, 2020.

[3] Shizhe Chen and others. Fine-grained Video-Text Retrieval with Hierarchical Graph Reasoning. CVPR, 2020.

---

### Decision · Program_Chairs · 2021-01-07
**Final Decision**

**Decision:**

Accept (Poster)

**Comment:**

This paper received 4 reviews with mixed initial ratings: 6,7,7,5. The main concerns of R1, who gave an unfavorable score, included lack of clarity (the manuscript is hard to follow) and limited empirical evaluation (the method is tested on a single synthetic dataset, CLEVRER). The latter point is echoed in other reviews as well. In response to that, the authors submitted a new revision and provided detailed responses to each of the reviews separately, which seemed to have addressed these concerns. R1 upgraded the rating and recommended acceptance.
As a result, the final recommendation is to accept this submission for presentation at ICLR as a poster.